# GEAR: A General Evaluation Framework for Abductive Reasoning

## Abstract

Since the advent of Large Language Models (LLMs), research has primarily focused on improving their instruction-following and deductive reasoning abilities. Yet a central question remains: can these models truly discover new knowledge, and how can we evaluate this ability? In this work, we address this gap by studying abductive reasoning—the process of generating plausible hypotheses to explain observations. We introduce **G**eneral **E**valuation for **A**bductive **R**easoning (GEAR), a new general-purpose, fully automated, transparent, and label-free evaluation paradigm that overcomes limitations of prior approaches. GEAR evaluates a set of hypotheses using three metrics: **consistency** (each hypothesis correctly explains the given observations), **generalizability** (consistent hypotheses make meaningful predictions on unseen inputs), and **diversity** (the set of hypotheses covers many distinct predictions and patterns). Built this way, GEAR is scalable (no human gold answers needed), reliable (transparent, deterministic scoring aligned with classical abduction), and open-ended (scores improve only when models produce new, plausible hypotheses, unlike existing static benchmarks that saturate once accuracy is high). Using GEAR, we conduct a fine-grained study of nine LLMs on four popular abduction benchmarks (1,500 problems), generating 50,340 candidate hypotheses. GEAR reveals model differences and insights that are obscured by prior gold-answer–based or purely human evaluations. We further propose a momentum-based curriculum training strategy that dynamically adjusts GEAR-derived training data by learning velocity: it begins with what the model learns faster and shifts toward harder objectives such as generating diverse hypotheses once the model is confident on foundational objectives (e.g., instruction following and consistency). Without gold-label supervision, this strategy improves all three GEAR objectives—consistency, generalizability, and diversity—and these gains transfer to established abductive-reasoning benchmarks. Taken together, GEAR provides a principled framework that not only evaluates abduction but also supplies label-free, scalable training signals that help LLMs produce more diverse and reliable hypotheses. We will release code and data upon acceptance.

## 1 Introduction

In the current AI community, there are many competing definitions of abductive reasoning. The most widely adopted one is Harman's view of abduction as *inference to the best explanation* (IBE) (Harman, 1965; Douven, 2021). Although this definition is simple and intuitive, it suffers from key limitations when applied to real-world settings, making benchmarks and evaluations built on it problematic.

**First**, IBE does not specify what counts as "best," and the criteria vary across contexts. In some cases, simplicity is prioritized; in others, novelty or explanatory power is preferred. As a result, IBE-based benchmarks often select a single "gold" hypothesis according to annotators' subjective judgments, yielding heterogeneous and unreliable labels (Zhao et al., 2023; Okasha, 2000; Cabrera, 2023). **Second**, multiple plausible hypotheses typically coexist. Real-world observations can often be explained in several ways, depending on how the data are conceptualized. This limitation appears across popular abductive benchmarks such as MINI-ARC (Kim et al., 2022), ACRE (Zhang et al.,

Figure 1: Underdetermination in abductive reasoning: a logically sound, observation-consistent hypothesis need not coincide with the annotated "gold" hypothesis; both can fit the seen data yet disagree on held-out predictions. Single-gold labeling can mask the plurality of valid explanations.

2021), LIST FUNCTIONS (Rule, 2020), and ARC-2025 (Chollet, 2019). For example, in Figure 1, the annotated "gold" hypothesis and other valid alternatives may all be well supported by sound abductive reasoning. However, existing evaluation usually excludes these alternatives by enforcing agreement with a single annotated "gold" hypothesis under a vague "best" standard. Philosophical accounts of abduction emphasize that hypotheses are often *evidentially underdetermined*: given finite data and background assumptions, more than one hypothesis may be well supported (Quine, 2014; Stanford, 2023), the ability to generate other plausible, well-supported hypotheses should also count as abductive success. Scientific progress relies on maintaining a diverse pool of feasible hypotheses. Although many will ultimately be falsified, they guide the design of experiments to discriminate, verify, or refute competing explanations, thereby enriching our knowledge. Even in current mature fields, credible alternative hypotheses often remain.

To address these limitations, we use Peirce's original definition of abduction, which frames it as a more general task of generating hypotheses from given observations (Frankfurt, 1958; Burks, 1946; Minnameier, 2004; Peirce, 1974). Building on this foundation, we propose GEAR, a new framework for systematically evaluating abductive reasoning. GEAR is grounded in three classical criteria for good scientific hypotheses: **(1) Consistency.** A hypothesis must not contradict observed facts, ensuring compatibility with existing evidence. **(2) Generalizability.** A good hypothesis extends beyond the observed data by making testable predictions on unseen cases. In Popper's terms, better hypotheses carry higher empirical content: they make riskier, more precise claims and thus invite more opportunities for refutation (Popper, 2005; 2014). We operationalize generalizability as the coverage of unseen inputs on which a hypothesis yields determinate predictions; larger coverage indicates greater generality. Accordingly, when two hypotheses fit the observed data, we prefer the broader (more falsifiable) one; if it withstands more severe tests, it is more strongly corroborated and more robust. **(3) Diversity.** A hypothesis should contribute a genuinely new perspective rather than echo existing ones, to avoid premature convergence on a single explanation. In the spirit of Chamberlin's multiple working hypotheses and Platt's strong inference (Chamberlin, 1965; Platt, 1964), we favor *sets of hypotheses* that articulate competing mechanisms testable by critical evidence. We quantify diversity with two complementary measures: $\gamma$-*diversity*, the average number of unique predictions per input across the hypotheses set (set-level variety), and $\beta$-*diversity*, the dissimilarity of prediction patterns between hypothesis pairs (dispersion). Higher diversity indicates broader causal coverage and multiple viewpoints on the observations; points of disagreement highlight decisive experiments and help accelerate scientific progress.

With GEAR, we re-examine four popularly used abduction benchmarks MINI-ARC, ACRE, LIST FUNCTIONS, and ARC-2025 across nine LLMs—spanning API-access models (GPT-o1 (OpenAI, 2024), GPT-4.1-mini (OpenAI, 2025a), GPT-o4-mini (OpenAI, 2025b)) and open-source models (LLAMA-3.3-70B, LLAMA-3.1-8B (Grattafiori et al., 2024), QWEN-2.5-72B, QWEN-2.5-7B (Qwen et al., 2025), GEMMA-2-9B (Team et al., 2024), NEXTCODER-7B (Aggarwal* et al., 2025))—and we find that— (1), Consistency remains hard, 70B-class models produce only 20% consistent hypotheses; (2) Consistency shows no significant correlation with the size of the initial observation set; (3), Model size is weakly related to abductive diversity—larger models do not necessarily generate more diverse hypotheses; and (4)existing gold-answer evaluations overlook the underdetermination inherent to abduction, around 80% of equally plausible hypotheses are labeled incorrect, and even the "accepted" hypotheses can differ substantially. Unlike prior frameworks that depend on gold answers or human raters, GEAR is label-free and fully automated, yielding dense, scalable signals that directly train models to generate consistent, generalizable, and diverse

hypothesis sets. We convert these signals into optimization targets for preference-based RL and fine-tune base models with LoRA, so that improvements on GEAR 's objectives are optimized end-to-end without gold supervision. To stabilize learning and broaden coverage across objectives, we introduce a momentum-based curriculum learning strategy that dynamically adjusts GEAR-derived training data by learning velocity: training begins with fast-to-learn, foundational objectives (instruction following and consistency) and shifts toward harder reasoning objectives that foster diverse hypothesis generation as competence increases. This procedure raises GEAR scores and transfers as accuracy gains on standard abductive-reasoning benchmarks across multiple model families (e.g., Qwen-2.5-7B, Llama-3.1-8B, NextCoder-7B).

## 2 RELATED WORK

**Reasoning: evolution from non-defeasible to defeasible.** Non-defeasible (deductive) reasoning preserves truth under added premises, whereas defeasible reasoning allows conclusions to be revised when new evidence appears (Yu et al., 2024). Abduction belongs to the latter: following Peirce, abduction proposes candidate hypotheses for observed facts, and deduction derives precise, testable predictions from a hypothesis (Frankfurt, 1958; Burks, 1946; Minnameier, 2004; Peirce, 1974). A key contrast with deduction is abduction's reliance on broad background knowledge (commonsense and domain-specific), which naturally yields multiple distinct yet plausible hypotheses for the same observation (He & Chen, 2025). Early symbolic systems struggled here due to narrow knowledge bases, whereas LLMs pretrained on large corpora make such abductive tasks more tractable (Yang et al., 2023; He & Chen, 2025). Despite its central role in discovery, abduction remains understudied compared with the extensive focus on deduction in AI (Niu et al., 2024; Liu et al., 2025; Yu et al., 2024; Huang & Chang, 2023; He & Chen, 2025).

**Gold- and human-based evaluation for abductive reasoning.** Current practice largely relies on two strands. *Gold answer-based* evaluation compares model outputs to a single reference either (i) at the *hypothesis level* using BLEU/ROUGE or embedding metrics such as BERTScore (Yang et al., 2024a; Qi et al., 2024; Movva et al., 2025; Bowen et al., 2024; Hua et al., 2025; Young et al., 2022), or (ii) at the *behavior level* by matching input–prediction pairs implied by the reference hypothesis (Sinha et al., 2019; Weston et al., 2015; Balepur et al., 2024; Shi et al., 2023; Wang et al., 2024; Rule, 2020; Chollet, 2019; Liu et al., 2024; Li et al., 2025; Chen et al., 2025; He et al., 2025). Despite scalability, single-reference matching is ill-suited to abduction: it rejects many plausible, logically sound alternatives that merely differ from the annotated answer; it is also costly (expert labeling) and unstable due to non-monotonic judgments and low inter-annotator agreement (Young et al., 2022). Complementarily, *human evaluation* is often used for qualities that are hard to algorithmically quantify (e.g., novelty, excitement) (Zhao et al., 2024; Qi et al., 2024; Yang et al., 2024b; Hu et al., 2024; Yang et al., 2025), but it is expensive, hard to reproduce or scale, and inherently subjective—particularly acute for abduction, where outcomes depend on rater expertise, instructions, and context, and small samples limit statistical power. In sum, both strands conflict with the essence of abductive reasoning: instead of testing agreement with a single "gold" explanation or subjective impressions, evaluations should assess a model's capacity to propose *multiple*, novel, and plausible explanatory hypotheses when underlying causes are unknown.

## 3 GEAR

We introduce the **G**eneral **E**valuation for **A**bductive **R**easoning (GEAR), an evaluation paradigm that scores hypotheses using *reference-free, transparent criteria* rather than agreement with a single gold answer. Unlike existing benchmarks that evaluate only a single generated hypothesis at a time, GEAR evaluates a *set* of hypotheses along three dimensions. An LLM exhibits stronger abductive proficiency when it can produce a hypothesis set that (i) correctly explains the given observations (**Consistency**), (ii) yields meaningful predictions on unseen inputs (**Generalizability**), and (iii) offers non-redundant alternatives rather than superficial variants (**Diversity**). Basic notation appears in Table 1.

Table 1: Abductive reasoning begins with a set of observations $\mathbb{O} = \{o_1, o_2, \dots\}$, where each $o_i := (\text{in}_i, \text{out}_i)$ is an input–output pair (e.g., $o_1 = (\text{floor}, \text{wet})$, $o_2 = (\text{air}, \text{humid})$). A hypothesis "it rained" can be represented as a function $f_{\text{rain}}$ mapping inputs to outputs, e.g., $f_{\text{rain}}(\text{floor}) = \text{wet}$, $f_{\text{rain}}(\text{sky}) = \text{cloudy}$. Each hypothesis $f$ has an input domain $\mathbb{D}$; outside this domain, predictions may be undefined or uninformative (e.g., $f_{\text{rain}}(\text{Math})$).

| Symbol | Meaning |
|---|---|
| $\mathbb{O} = \{o_1, o_2, \dots\}$ | Set of observations to be explained |
| $o_i := (\text{in}_i, \text{out}_i)$ | Observation as an input–output pair |
| $f$ | Hypothesis function |
| $\mathbb{F}$ | A set of hypotheses |
| $\mathbb{D}$ | (Effective) input domain of hypothesis $f$ |
| $M$ | A set-size measure (e.g., cardinality $|\cdot|$) |
| $M(\mathbb{D})$ | Size of the input domain $\mathbb{D}$ |
| $\tilde{f}$ | Trivial hypothesis that memorizes all seen observations |
| $\mathbb{S} = \{\text{in}_1, \text{in}_2, \dots\}$ | Problem-specific sample space of candidate inputs |
| $\mathbb{P}_f := \{(\text{in}, f(\text{in})) : \text{in} \in \mathbb{S}\}$ | Prediction space of $f$ on $\mathbb{S}$ |

## 3.1 CONSISTENCY

Consistency is the most fundamental requirement of a hypothesis: it must not conflict with observed facts. Formally, a generated hypothesis $f$ is consistent with the observation set $\mathbb{O}$ if $\forall (\text{in}_i, \text{out}_i) \in \mathbb{O}, f(\text{in}_i) = \text{out}_i$. This criterion guarantees agreement with all known observations and underlies most existing evaluations of hypothesis generation, including gold answer–based evaluations.

## 3.2 GENERALIZABILITY

Given several consistent hypotheses, a more general hypothesis is one that yields predictions on a broader set of unseen cases. A more general hypothesis confers two advantages: (1) it can be applied in more future situations, increasing its practical utility; and (2) because it makes predictions for more situations, it can be tested more extensively and—if it survives—becomes correspondingly more robust (Popper, 2005; 2014). Formally, for two hypotheses $f_1$ and $f_2$ with respective input domains $\mathbb{D}_1$ and $\mathbb{D}_2$ and a set-size measure $M$, if $M(\mathbb{D}_1) > M(\mathbb{D}_2)$, then $f_1$ is considered more general than $f_2$, with a simple example.

For example, given three observations $o_1 = (1, 1)$, $o_2 = (10, 1)$, and $o_3 = (100, 1)$, a trivial lookup-table hypothesis $\tilde{f}$ that merely memorizes these pairs is consistent yet fails to generalize. In contrast, $f_1(n) = \text{rev}(n)$ (digit-reversal on integers), $f_2(x) = x/x$ for $x \neq 0$ (undefined at $x = 0$), and the constant hypothesis $f_3(x) \equiv 1$ are all consistent but differ in generalizability. Under the simple size measure $M(\mathbb{D}) = |\mathbb{D}|$, we have: $\{1, 10, 100\} \subset \mathbb{N}_0 \subset \mathbb{R} \setminus \{0\} \subset \mathbb{R} \Rightarrow \tilde{f} \prec f_1 \prec f_2 \prec f_3$.

The effective domain $\mathbb{D}$ and its size measurement $M$ are not static but depend on the problem context and representation. For instance, in arithmetic, $f_1$ (reversal) is defined on integers, whereas in programming tasks the same hypothesis applies to strings, lists, and other finite sequences, making $f_1$ more general than $f_2$ in that setting. Since it is generally infeasible to determine a global domain $\mathbb{D}$ across all conceivable contexts, GEAR evaluates generalizability relative to a pre-defined *problem-specific sample space* $\mathbb{S}$ shared across all hypotheses under comparison, together with its associated measurement $M$. This sample space serves as the operational domain for evaluation and provides the basis for comparing the generalizability of different hypotheses.

## 3.3 DIVERSITY

Prior work typically measures hypothesis diversity with black-box text similarity metrics such as BERTScore (Zhang et al., 2019)—which capture surface-level semantic overlap rather than underlying explanatory mechanisms—or with subjective, annotator-based judgments. In contrast, following the Multiple Working Hypotheses view (Chamberlin, 1965; Platt, 1964), we define diversity directly from *prediction patterns*: distinct hypotheses should be *separable* by some input in $\mathbb{S}$. Concretely, for two consistent hypotheses $f_1, f_2$ and an unseen input $\text{in}_x \in \mathbb{S}$, if $f_1(\text{in}_x) \neq f_2(\text{in}_x)$, then $f_1$ and $f_2$ are separable on $\text{in}_x$; the more such inputs exist, the more diverse the hypotheses are.

In light of classical ecological diversity theory—specifically $\beta$- and $\gamma$-diversity (Whittaker, 1960)—we adapt set-based diversity ideas to hypothesis sets, not as a one-to-one import from ecol-

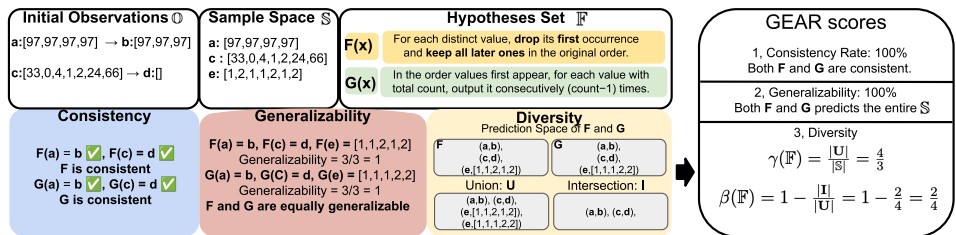

Figure 2: GEAR with a live example with $|\mathbb{F}| = 2, |\mathbb{S}| = 3$.

ogy but as a structural analogy over prediction patterns. Concretely, we define $\gamma$ as the average number of unique predictions per input over a set of hypotheses $\mathbb{F}$ and $\beta$ as the mean pairwise Jaccard dissimilarity between prediction sets on a shared sample space.

$\gamma$-**diversity** (*average unique predictions per input*). Let $\mathbb{P}_f = \{(\text{in}, f(\text{in})) : \text{in} \in \mathbb{S}\}$. Define

$$\gamma(\mathbb{F}; \mathbb{S}) \;:=\; \frac{1}{M_1(\mathbb{S})} \, M_2\!\left( \bigcup_{f \in \mathbb{F}} \mathbb{P}_f \right) \;=\; \frac{1}{|\mathbb{S}|} \sum_{\text{in} \in \mathbb{S}} \Big| \big\{ (\text{in}, f(\text{in})) : f \in \mathbb{F} \big\} \Big|,$$

Under cardinality $M_1 = M_2 = |\cdot|$ and $|\mathbb{F}| = m$, $\gamma \in [1, m]$, A hypothesis proposer that generates near-duplicate hypotheses will yield $\gamma \approx 1$ because most hypotheses agree on almost all inputs. Conversely, a proposer that views the observations from diverse perspectives and produces genuinely novel hypotheses will achieve a larger $\gamma$, approaching $\gamma \approx m$ when predictions are mutually distinct for every input. Notably, $M_1$ is the size measure on the sample (input) space, whereas $M_2$ is the size measure on the prediction space; the two need not be identical.

$\beta$-**diversity** (*prediction-pattern dispersion*). Measure pairwise dispersion via the Jaccard *dissimilarity* between prediction sets, and then average across all pairs:

$$d_J(f_i, f_j) \;:=\; 1 - \frac{M\big(\mathbb{P}_{f_i} \cap \mathbb{P}_{f_j}\big)}{M\big(\mathbb{P}_{f_i} \cup \mathbb{P}_{f_j}\big)}, \qquad \beta(\mathbb{F}; \mathbb{S}) \;:=\; \frac{2}{n(n-1)} \sum_{1 \le i < j \le m} d_J(f_i, f_j).$$

**Example.** Let $\mathbb{S} = \{0, 1, 2\}$, $\mathbb{P}_{f_1} = \{(0,1), (1,2), (2,3)\}$, $\mathbb{P}_{f_2} = \{(0,1), (1,2), (2,2)\}$, and $M = |\cdot|$. *Generalizability:* for each $f$, $G(f) = |\mathbb{P}_f|/|\mathbb{S}|$. Thus $G(f_1) = G(f_2) = 3/3 = 1$. *Set coverage:* $\bigcup_i \mathbb{P}_{f_i} = \{(0,1), (1,2), (2,2), (2,3)\}$, so $\gamma = \big| \cup_i \mathbb{P}_{f_i} \big|/|\mathbb{S}| = 4/3$. *Diversity:* the intersection size is 2 and the union size is 4, hence the Jaccard distance $d_J(f_1, f_2) = 1 - \frac{2}{4} = \frac{1}{2}$; with a single pair, $\beta = \frac{1}{2}$. (If fewer than two consistent hypotheses are generated, we set $\beta = 0$.) An additional example from LIST FUNCTIONS is shown in Fig. 2.

While $\gamma$ and $\beta$ are mathematically related, neither uniquely determines the other. The formal relationship and proofs are provided in Appendix F.

Because Consistency, Generalizability, and Diversity have precise mathematical definitions, they can be computed directly—without human labor or auxiliary black-box models—at evaluation time. Consequently, GEAR is (1) *scalable*: it runs automatically on any newly generated hypotheses; (2) *reliable*: all metrics are transparently defined and computed, including Diversity, which GEAR measures by underlying mechanisms rather than via black-box proxies; and (3) *open-ended*: GEAR evaluates how many genuinely different explanatory perspectives a model can produce and, because the hypothesis space is in principle unbounded, it imposes no upper limit on valid hypotheses.

## 4 LLM EVALUATION SETTINGS

**Benchmarks.** We use four widely used abductive benchmarks: MINI-ARC (Kim et al., 2022), ACRE (Zhang et al., 2021), LIST FUNCTIONS (Rule, 2020), and ARC-2025 (Chollet, 2019). These gold-answer benchmarks split observations into $\mathbb{O}_{\text{train}}$ and $\mathbb{O}_{\text{test}}$. Unlike the traditional setting—where models see only $\mathbb{O}_{\text{train}}$ and are judged on $\mathbb{O}_{\text{test}}$—GEAR measures *how many* consistent hypotheses a model can produce and *how diverse* they are. Because several datasets provide only a few observations per problem (e.g., 2–3 train and 1 test), we pool all observations into $\mathbb{O}_{\text{all}} := \mathbb{O}_{\text{train}} \cup \mathbb{O}_{\text{test}}$ to enable broader analyses. We choose these datasets because (i) hypotheses are expressible in formal languages (e.g., executable programs), allowing deterministic evaluation

without extra adaptation; and (ii) each dataset provides sufficiently diverse observations to seed abduction.

**Sampling initial observations.** We study the effect of observation size by using $n \in \{1, 2, 3, 4\}$ observations. For ACRE, outputs are Boolean (`on`/`off`); with a single observation, label semantics may be ambiguous, so we use $n \in \{2, 3, 4\}$ and ensure at least one `on` and one `off`. When $n$ observations are required, we sample $n$ pairs from $\mathbb{O}_{\text{all}}$ without replacement to form $\mathbb{O}_n$. We randomly select 100 problems per dataset. For three datasets we use $n = 4$ (total $3 \times 100 \times 4 = 1200$), and for ACRE we use $n = 3$ (total $1 \times 100 \times 3 = 300$), yielding 1,500 problems overall.

**Generation protocol.** Given $\mathbb{O}_n$, we prompt with a dataset-agnostic template $P_{\text{init}}$ to obtain the first hypothesis $f_1$. We then iterate with $P_{\text{iter}}$, which lists previously generated hypotheses $\mathbb{F}_{t-1} = \{f_1, \ldots, f_{t-1}\}$ and requests a *new* $f_t$ that is (i) consistent with $\mathbb{O}_n$ and (ii) distinct from all $f \in \mathbb{F}_{t-1}$. Both $P_{\text{init}}$ and $P_{\text{iter}}$ are shared across datasets (see $P_{\text{init}}$ and $P_{\text{iter}}$ in the Appendix G).

**Stopping rule.** Since it is infeasible to enumerate all hypotheses an LLM could generate for a given problem, we use a *quality-triggered* early stop: generation for a problem halts once the model emits three "bad" hypotheses—unparseable, inconsistent with $\mathbb{O}_n$, or non-novel w.r.t. $\mathbb{F}_{t-1}$. This keeps runs finite and comparable while allowing stronger models to produce more consistent, novel hypotheses before stopping. Formal criteria and thresholds are in Appendix B.

**Sample space $\mathbb{S}$.** We assess generalizability and diversity on a fixed, dataset-specific $\mathbb{S}$. Since exhaustive coverage is infeasible, we construct $\mathbb{S}$ via simple, reproducible rules with a fixed seed. For LIST FUNCTIONS and ACRE, whose input domains are explicit (lists; entity triples), we stratify by input size: include the empty case, include all singletons, and uniformly subsample the remainder. For MINI-ARC/ARC-2025, naive enumeration of visual grids mostly yields noise; thus $\mathbb{S}$ is the set of unique input grids from the official splits after canonicalization and deduplication. Full construction details and cardinalities are in Appendix A.

# 5 ANALYSIS OF LLM EVALUATIONS

## 5.1 MAIN ANALYSIS: LLM PERFORMANCE UNDER GEAR

Across nine LLMs we collect 50,340 hypotheses, with 17,835 *consistent* and 32,505 *inconsistent*; of the latter, 4,346 fail format/parseability (e.g., non-executable) and 28,159 contradict at least one observation (Fig. 3).

**Overall volume and consistency.** Under the quality-triggered early stop, the hypotheses-per-problem reflect overall generation capacity. GPT-O4-MINI and GPT-O1 produce the most consistent hypotheses and the highest consistency rates, yielding $\sim 2$–$4$ more hypotheses per problem than the next tier (panel (a)) and correspondingly higher consistency (panel (c)).

**Effect of initial observation size.** Increasing I/O pairs from 1 to 4 tightens constraints: both $\beta$ and $\gamma$ diversity decline (panels (d)–(e)), while the consistency rate remains comparatively stable (panel (c)). Early stopping is increasingly triggered by the novelty threshold rather than inconsistency.

**Instruction following.** Most models adhere to the required format (panel (b)); LLAMA-3.1-8B is an outlier, often appending free-form text or violating the code template, harming parseability. (§6 shows RL substantially closes this gap.)

**Model size vs. abductive diversity.** Among open-source models, parameter count correlates weakly with abductive diversity: LLAMA-3.3-70B trails GEMMA-2-9B and QWEN-2.5-7B on $\beta$-diversity (panel (d)), and QWEN-2.5-72B is only on par with these smaller models—suggesting data and training procedure may matter more than raw size for abductive reasoning.

**Generalizability.** Because $\mathbb{S}$ pertains to each dataset's distribution, consistent hypotheses achieve high coverage ($\approx 95\%$ on average; panel (f)). Corner cases remain in MINI-ARC/ARC-2025 (e.g., infinite loops). Since GPT-O1 and GPT-O4-MINI contribute a large share of consistent hypotheses on these harder datasets, their macro-averaged coverage appears lower—not due to intrinsically weaker generality, but because the solved problems are more challenging.

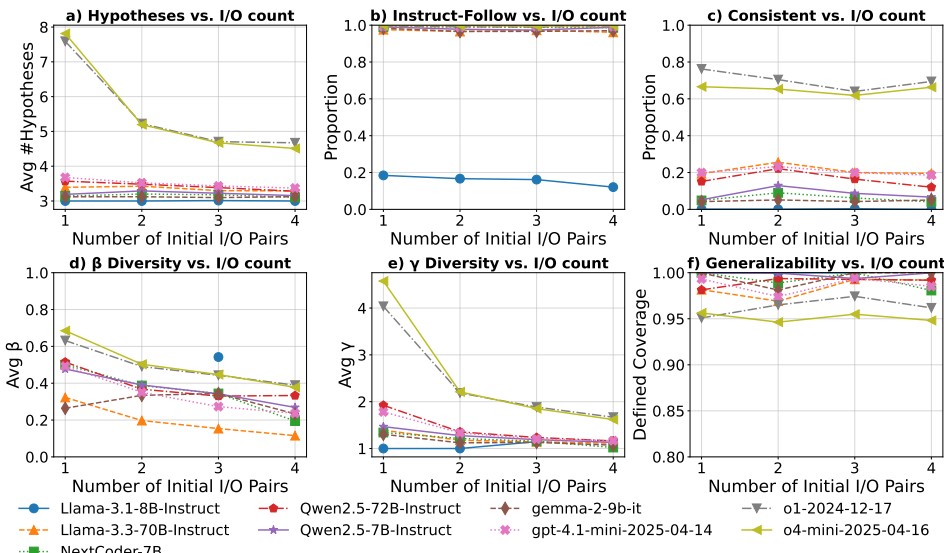

Figure 3: Model performance under GEAR. Metrics are *macro-averaged per problem*: compute the value per problem, then take the unweighted mean. Panels (a)–(c) use *all* generated hypotheses; panels (d)–(f) restrict to *consistent* hypotheses, since diversity and generalizability are only meaningful for consistent sets.

## 5.2 SIMULATION STUDY 1: ABDUCTIVE REASONING IS DEFEASIBLE

As noted in § 2, abduction is *defeasible*: a hypothesis generated by logical and sound abductive reasoning need not coincide with the "gold answer." Benchmarks that enforce a single gold hypothesis thus reward label matching rather than generating alternative, plausible explanations; and we also show that without extensive test cases previous benchmarks *cannot* even explicitly distinguish an alternative hypothesis from the gold-labeled one as intended.

**Empirical illustration.** We previously generated 17,835 hypotheses consistent with initial observations $\mathbb{O}_n$ ($n \in \{1, 2, 3, 4\}$, sampled from $\mathbb{O}_{\text{all}}$). For each problem, fixing $\mathbb{O}_n$ and its consistent hypotheses set $\mathbb{F}$, we sample $m \in \{1, 2, 3, 4\}$ hidden test pairs $\mathbb{O}_m \subset \mathbb{O}_{\text{all}} \setminus \mathbb{O}_n$ and repeat each $(n, m)$ five times. A hypothesis *passes* if it agrees with all test cases in $\mathbb{O}_m$. We report (a) pass rate and (b) $\gamma$-diversity of survivors, averaged over problems (Fig. 4a–b).

Pass rates decline as $m$ increases but do not collapse; even at $m=4$, a nontrivial fraction remains valid. Survivors also retain diversity—for example, on ACRE at $m=4$ the passed set has $\gamma \approx 1.5$, i.e., roughly 1.5 distinct predictions per input on the shared sample space. Because all candidates are LLM-generated, this is a conservative lower bound on plausible explanations. As available observations and background knowledge broaden, underdetermination grows, and single-gold-answer metrics increasingly misclassify reasonable alternatives.

## 5.3 SIMULATION STUDY 2: GEAR IS GENERAL

In this section, we show that GEAR is **general** rather than task-specific. Across varied abductive datasets, higher GEAR scores predict a greater chance that the set contains a hypothesis explaining unseen (hidden) observations, thereby aligning with existing evaluations of good abduction and indicating cross-task applicability.

For each problem, we sample $m \in \{1, 2, 3, 4\}$ hidden cases $\mathbb{O}_m \subset \mathbb{O}_{\text{all}} \setminus \mathbb{O}_n$ and draw a context $\mathbb{F}_c$ of size $c \in \{0, 1, 2\}$ from hypotheses consistent with $\mathbb{O}_n$; we discard any context with a member already passing $\mathbb{O}_m$. Let $\mathbb{F}_{\text{consistent}}$ be the full set of hypotheses consistent with $\mathbb{O}_n$ and from the remaining consistent pool $\mathbb{F}_{\text{consistent}} \setminus \mathbb{F}_c$, we enumerate unordered pairs $(f_a, f_b)$ and, for each $f \in \{f_a, f_b\}$, compute marginal diversity gain for $\rho \in \{\gamma, \beta\}$ as $\Delta_\rho(f) = \rho(\mathbb{F}_c \cup \{f\}; \mathbb{S}) - \rho(\mathbb{F}_c; \mathbb{S})$, define generalizability as $g(f) = \frac{|\mathbb{P}_f|}{|\mathbb{S}|}$, the coverage of $f$ over the sample space $\mathbb{S}$, and an average GEAR score: $\text{Score}(f) = \frac{1}{3}\big(g(f) + \Delta_\gamma(f) + \Delta_\beta(f)\big)$. We label $(f_{\text{chosen}}, f_{\text{rejected}})$ so that $\text{Score}(f_{\text{chosen}}) > \text{Score}(f_{\text{rejected}})$. We then test both on $\mathbb{O}_m$, declare a pass when a hypothesis explains all hidden cases, and aggregate over problems/contexts/pairs to report the odds ratio $\text{OR}_m = \Pr(\text{pass} \mid f_{\text{chosen}}) / \Pr(\text{pass} \mid f_{\text{rejected}})$.

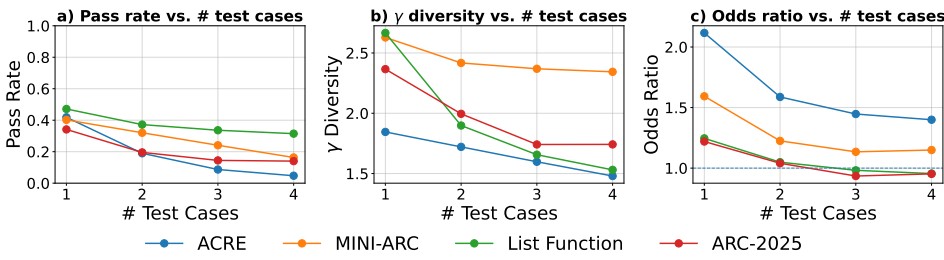

Figure 4: Simulation Study Results

Figure 4(c) shows that $\mathrm{OR}_m$ tends toward 1 as $m$ increases, yet in most settings $\mathrm{OR}_m > 1$: the higher-GEAR-score candidate is more likely to pass the hidden cases, aligning with the idea that a more diverse prediction space on $\mathbb{S}$ increases the chance of covering the unseen evidence $\mathbb{O}_m$.

# 6 REINFORCEMENT LEARNING WITH GEAR

## 6.1 TRAINING DATA PREPARATION

Building on the simulation study above, we posit that learning *better abduction*—operationalized as generating a hypothesis set that aligns better with GEAR —which naturally improves the chance that at least one hypothesis fits unseen observations and thereby *generalizes* to downstream tasks, without requiring any gold hypotheses as supervision.

Training data are constructed from the previously generated $50,340$ hypotheses (§5.1). Within each dataset (100 problems), we sample 50 for training, 10 for validation, and hold out 40 for final evaluation. Similar to the preference-pair construction in § 5.3, for each training problem, let $\mathbb{F}_{\text{all}}$ be the set of hypotheses generated by nine LLMs (including both inconsistent and consistent hypotheses), and let $\mathbb{F}_c \subseteq \mathbb{F}$ be a small hypothesis context of size $c \in \{0, 1, 2\}$ sampled from $\mathbb{F}_c \subseteq \mathbb{F}_{\text{consistent}}$. We then enumerate all unordered hypothesis pairs $(f_a, f_b)$ from $\mathbb{F}_{\text{all}} \setminus \mathbb{F}_c$. For each $f \in \{f_a, f_b\}$, we assign a preference in three stages: (1) *instruction following / format compliance* (prefer parsable over non-parsable outputs; $159,981$ pairs); (2) *consistency* (among parsable candidates, prefer those consistent with $\mathbb{O}_n$; $429,980$ pairs); (3) when both are parsable and consistent, GEAR score, where we score each candidate by the same score function $\text{Score}(f)$ in § 5.3. We prefer the candidate with the higher $\text{Score}(f)$ ($249,561$ pairs). In total, this yields $839,522$ training preference pairs. We fine-tune the base models with Direct Preference Optimization (DPO) using LoRA (rank 128, $\alpha = 256$) under 4-bit quantization with bfloat16 compute. For DPO, we randomly sample $51,200$ pairs with a fixed 1:1:1 ratio across Parsing, Consistency, and GEAR preferences.

## 6.2 MOMENTUM-BASED CURRICULUM LEARNING

During fine-tuning we observed that the *mixture* of preference data materially affects outcomes, and different base models favor different mixtures (some benefit from earlier emphasis on format/consistency, others from earlier diversity). This led us to a momentum-based curriculum method. The intuition is to learn what improves *fastest* and is *easiest* first, then shift weight toward harder signals. We did an ablation study on the effect of each preference category (See in Appendix E). This adaptively reweights based on measured learning progress, avoiding hand-tuned fixed ratios and letting each base model gravitate to its preferred mixture.

**Adaptive reweighting preferences.** Unlike existing curriculum learning methods that either prescribe a static training curriculum—thereby overlooking model-specific competence differences (Bengio et al., 2009; Wang et al., 2019)—or adopt dynamic schemes that operate at the sample level (Zhou et al., 2021; Jiang et al., 2015; Sow et al., 2025), which are computationally expensive and often misaligned with the higher-level objective of producing diverse hypotheses, our approach is simple yet efficient: it dynamically delivers a skill-level curriculum—i.e., a goal-aligned schedule over preference types (Parsing/Format, Consistency, and GEAR/diversity) rather than over individual instances—aligned with GEAR objectives.

For each preference type $r$, keep an Exponential Weighted Moving Average (EWMA) of its probe loss and convert recent improvement into sampling weight:

$$E_r^{(t)} = (1 - \alpha)\, E_r^{(t-1)} + \alpha\, L_r^{(t)}, \quad m_r^{(t)} = \text{clip}\big(E_r^{(t-1)} - E_r^{(t)},\, 0,\, m_{\max}\big), \quad p_r^{(t)} \;\propto\; \varepsilon + m_r^{(t)}$$

Table 2: Aggregated results across nine settings.

GEAR *performance improvement*

| | Instruction Following Rate | Consistency | Generalizability | $\beta$ | $\gamma$ |
|---|---|---|---|---|---|
| Llama-3.1-8b | 0.149 | 0.002 | 1.000 | 0.000 | 1.000 |
| Llama-3.1-8b-dpo-fixed-ratio | 0.968 | 0.014 | 1.000 | 0.000 | 1.000 |
| Llama-3.1-8b-dpo-momentum-curriculum | **0.970** | **0.020** | **1.000** | **0.196** | **1.179** |
| qwen-2.5-7b | 0.988 | 0.037 | 0.987 | 0.032 | 1.001 |
| qwen-2.5-7b-dpo-fixed-ratio | 0.996 | **0.042** | 0.987 | 0.044 | 1.046 |
| qwen-2.5-7b-dpo-momentum-curriculum | **0.997** | 0.041 | **1.000** | **0.073** | **1.052** |
| nextcoder-7b | 0.863 | 0.013 | 1.000 | 0.140 | 1.081 |
| nextcoder-7b-dpo-fixed-ratio | 0.965 | 0.028 | **1.000** | 0.078 | 1.066 |
| nextcoder-7b-momentum-curriculum | **0.991** | **0.030** | 0.991 | **0.182** | **1.151** |

*Cross-task generalization with* GEAR

| | Avg Train Pass Rate | Avg Test Pass Rate | Top-1 Acc. | Top-2 Acc. | Top-3 Acc. |
|---|---|---|---|---|---|
| Llama-3.1-8b | 0.011 | 0.009 | 0.000 | 0.003 | 0.004 |
| Llama-3.1-8b-dpo-fixed-ratio | 0.096 | 0.076 | 0.028 | 0.043 | 0.049 |
| Llama-3.1-8b-dpo-momentum-curriculum | **0.129** | **0.101** | **0.035** | **0.052** | **0.065** |
| qwen-2.5-7b | 0.198 | 0.154 | 0.035 | 0.059 | 0.068 |
| qwen-2.5-7b-dpo-fixed-ratio | **0.205** | 0.160 | 0.054 | 0.064 | 0.077 |
| qwen-2.5-7b-dpo-momentum-curriculum | 0.203 | **0.163** | **0.066** | **0.083** | **0.095** |
| nextcoder-7b | 0.151 | 0.121 | 0.025 | 0.044 | 0.055 |
| nextcoder-7b-dpo-fixed-ratio | 0.173 | 0.136 | 0.051 | 0.065 | 0.074 |
| nextcoder-7b-momentum-curriculum | **0.189** | **0.149** | **0.065** | **0.080** | **0.092** |

*Where* $r \in \mathcal{R}$ indexes preference types; $L_r^{(t)}$ is the probe loss on the validation subset at epoch $t$; $E_r^{(t)}$ is its EWMA; $\alpha \in (0,1)$ is the smoothing factor; $m_r^{(t)}$ is the clipped improvement (capped by $m_{\max} > 0$); $p_r^{(t)}$ is the per-type sampling weight (normalized across $r$); $\varepsilon > 0$ prevents zero weight; and $p_r^{(t)}$ is then normalized across $r$ and clipped to $[w_{\min}, w_{\max}]$.

*Experimental settings.* We use $\alpha = 0.1$, $\varepsilon = 0.1$, $m_{\max} = 0.03$, $w_{\min} = 0.8$, $w_{\max} = 1.2$, and update the sampling weights every 1,280 training examples. The same hyperparameters and schedule were used for all three fine-tuned models (no per-model tuning).

For evaluation, we use the original (non-sampled) splits with $\mathbb{O}_{\text{train}}$ and $\mathbb{O}_{\text{test}}$ from held-out problems, asking each model to generate *three* hypotheses per problem. As shown in Table 2, models trained from these GEAR-derived preferences—despite never seeing held out problems—achieve higher diversity scores ($\beta, \gamma$) and improved Top-1/2/3 (T3) accuracies, with the momentum-based curriculum consistently outperforming the fixed-ratio baseline across the reported settings.

# 7 CONCLUSION & DISCUSSION

We present GEAR, a general evaluation framework for abduction that scores hypotheses by consistency, generalizability, and diversity. Across nine LLMs on four abduction benchmarks, GEAR reveals differences that gold- or human-centric evaluations miss; simulation study confirms abduction is defeasible and shows that more diverse hypothesis sets are more likely to predict hidden observations. We convert GEAR into label-free training signals and propose a momentum-based DPO curriculum that adapts preference weights with learning progress, improving hypothesis diversity and downstream accuracy without gold supervision.

Although our experiments use programmable domains, GEAR is not limited to them. The framework hinges on four ingredients: (i) a size measure $M$, (ii) a sample space $\mathbb{S}$, (iii) a deduction/execution oracle, and (iv) a semantic equivalence predicate. In natural-language (NL) settings these remain the same but grow harder: $M$ should capture semantic coverage (e.g., topical/diversity weightings) rather than raw cardinality; $\mathbb{S}$ can be unlabeled yet must include many diverse probes to reveal prediction patterns; the deduction step becomes model- or tool-mediated and thus stochastic, mitigated by calibrated decoding, tool use, and repeated sampling; and equivalence must be judged semantically or via canonicalization to executable meaning representations to curb polysemy. The primary bottleneck is therefore foundational NL tooling for reliable execution and equivalence under ambiguity. GEAR 's applicability to NL tasks scales with the quality of these primitives; as they improve, the framework transfers with minimal changes—largely a swap of stronger semantic metrics and oracles.

## ETHICS STATEMENT

We affirm adherence to the ICLR Code of Ethics. This work evaluates and trains language models on publicly available benchmarks (MINI-ARC, ACRE, LIST FUNCTIONS, ARC-2025) and model APIs; it does not involve human subjects, personally identifiable information, or sensitive attributes. To minimize potential harms, we restrict experiments to benign, programmable tasks. Dataset licenses and usage follow their original terms; no proprietary or private data are redistributed. There are no known conflicts of interest or third-party sponsorships that influenced results; any such relationships will be disclosed upon de-anonymization. Code and data used for evaluation will be released to facilitate auditing and responsible reuse.

## REPRODUCIBILITY STATEMENT

We aim for full reproducibility. Datasets, problem sampling, and the construction of the evaluation sample spaces $S$ are defined by simple, deterministic rules with fixed seeds; prompts for hypothesis generation (initial/iterative) are provided; and all scoring criteria (Consistency, Generalizability, $\beta/\gamma$ Diversity) are formally specified. Training details (preference construction, DPO with LoRA, quantization, and the momentum-based curriculum schedule with hyperparameters) are described alongside exact settings. We will release code, prompts, and configuration files enabling end-to-end replication—from hypothesis generation to metric computation and fine-tuning—together with random seeds and logs upon acceptance.

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

## A  Sample space $\mathbb{S}$

The sample space $\mathbb{S}$ is central to GEAR: a broader $\mathbb{S}$ exposes more inputs on which prediction patterns can be compared, thus providing a sharper lens for assessing generalizability and diversity. In practice, however, $\mathbb{S}$ must balance breadth with computational cost and with the effort required to construct valid, meaningful inputs. We therefore define $\mathbb{S}$ per dataset using simple, reproducible rules and a fixed random. And for simplicity, we use cardinality $(|\cdot|)$ as set-size measurement in our evaluation.

**List Functions:** In the original dataset, Inputs are lists of integers with element domain $\{0, \ldots, 99\}$ and length $k \in \{0, \ldots, 15\}$. After expansion, the full combinatorial space has size $\sum_{k=0}^{15} 100^k$, which is infeasible to exhaust at evaluation time. We adopt *stratified sampling by length*: (i) include the unique empty list for $k{=}0$; (ii) include all 100 singletons for $k{=}1$; (iii) for each $k \in \{2, \ldots, 15\}$, uniformly sample (without replacement) up to 1,000 lists from the $100^k$ possibilities. This yields $|\mathbb{S}_{\text{ListFunc}}| = 1 + 100 + 14 \times 1{,}000 = 14{,}101$. (*One could broaden the input domain beyond the original dataset—for example by allowing negative or floating-point values, or even arbitrary real numbers—but we retain the original integer domain; in our experiments this range is already sufficient to probe diverse hypothesis behaviors.*)

**ACRE:** In the original dataset, each primitive entity is a triple $\langle \text{color}, \text{shape}, \text{material} \rangle$ with color $\in \{\text{blue}, \text{brown}, \text{cyan}, \text{gray}, \text{green}, \text{purple}, \text{red}, \text{yellow}\}$ (8), shape $\in \{\text{cube}, \text{cylinder}, \text{sphere}\}$ (3), material $\in \{\text{metal}, \text{rubber}\}$ (2), so the vocabulary has $8 \times 3 \times 2 = 48$ distinct entity types. An input is an *ordered* list (repetitions allowed) of $c$ entities with $c \in \{0, \ldots, 8\}$ since in the original dataset at most 8 entities are seen in one observation. We again use stratified sampling by $c$: include the empty list for $c{=}0$; include all 48 singletons for $c{=}1$; for each $c \in \{2, \ldots, 7\}$, uniformly sample up to 1,000 lists without replacement. This gives $|\mathbb{S}_{\text{ACRE}}| = 1 + 48 + 7 \times 1{,}000 = 7{,}049$. (*Broader stress tests are possible by extending the vocabulary with new colors, shapes, or materials, but we restrict ourselves to the original schema here; empirically this space is already rich enough to discriminate among hypotheses.*)

**MINI-ARC and ARC-2025:** Unlike the previous two datasets, these benchmarks encode inputs as small integer grids (values in a fixed palette) that carry *visual semantics*. Naively enumerating grids within a size range produces overwhelmingly non-meaningful noise and is therefore counterproductive for abductive reasoning. Instead, we define $\mathbb{S}$ as the set of all *unique* input grids already present in the official splits (train/validation/test), after canonical serialization and deduplication of exact matches. This pragmatic choice keeps inputs semantically meaningful while avoiding sampling artifacts. It yields $|\mathbb{S}_{\text{MINIARC}}| = 767$, $|\mathbb{S}_{\text{ARC2025}}| = 4{,}826$.

**Reproducibility:** All sampling steps use a fixed random seed and uniform sampling without replacement within each stratum; we publish the materialized $\mathbb{S}$ for each dataset to ensure exact reproducibility of scores.

## B  Stopping rule and "bad" hypotheses.

Enumerating all potential hypotheses a model could generate is infeasible. To keep generation finite and comparable across models, we adopt a *quality-triggered* early-stopping rule: generation for a problem stops once the model accumulates three "bad" hypotheses. A hypothesis is *bad* if it satisfies any of the following: (i) it cannot be parsed into executable code (format or syntax error); (ii) it is inconsistent with $\mathbb{O}_n$ (violates at least one given observation); or (iii) it lacks novelty relative to prior hypotheses. For (iii), let the shared sample space be $\mathbb{S}$ and define:

$$\text{cov}\big(f \mid \mathbb{F}_{t-1}\big) := \frac{1}{|\mathbb{S}|} \Big|\{\text{in} \in \mathbb{S} : \exists g \in \mathbb{F}_{t-1} \text{ s.t. } g(\text{in}) = f(\text{in})\}\Big|$$

We mark $f$ as non-novel if $\text{cov}(f \mid \mathbb{F}_{t-1}) \geq 0.8$ (i.e., at least 80% of its predictions on $\mathbb{S}$ are duplicate relative to $\mathbb{F}_{t-1}$). Compared with a hard quota on the number of hypotheses, this early-stopping rule (i) prevents unbounded generation and degenerate cycling; (ii) allows stronger models to produce more consistent and novel hypotheses before stopping; and (iii) reduces the chance that a fixed quota artificially limits performance.

## C TABLES

Table 3: Per-dataset GEAR scores of nine LLMs (Instruction Following, Consistency, $\gamma$, $\beta$, Generalizability).

| Model | Dataset | IF | Cons. | Norm-$\gamma$ (q=0) | $\beta$-Struct | Gen. |
|---|---|---|---|---|---|---|
| o4-mini-2025-04-16 | Avg | 0.9972 | 0.6708 | 2.3330 | 0.5036 | 0.9398 |
| | ACRE | 1.0000 | 0.9771 | 1.6718 | 0.4774 | 0.9816 |
| | MINI-ARC | 0.9958 | 0.4489 | 2.1715 | 0.4850 | 0.9472 |
| | List Fns | 0.9978 | 0.9449 | 3.4245 | 0.4850 | 0.9727 |
| | ARC-2025 | 0.9951 | 0.3121 | 2.0644 | 0.5670 | 0.8577 |
| gpt-4.1-mini-2025-04-14 | Avg | 0.9911 | 0.2263 | 1.3320 | 0.3457 | 0.9824 |
| | ACRE | 0.9926 | 0.5298 | 1.2754 | 0.3402 | 0.9863 |
| | MINI-ARC | 0.9877 | 0.0622 | 1.3708 | 0.3710 | 1.0000 |
| | List Fns | 0.9917 | 0.2827 | 1.4464 | 0.3001 | 0.9859 |
| | ARC-2025 | 0.9925 | 0.0304 | 1.2355 | 0.3716 | 0.9575 |
| o1-2024-12-17 | Avg | 0.9924 | 0.7165 | 2.2677 | 0.4892 | 0.9538 |
| | ACRE | 0.9888 | 0.9679 | 1.6813 | 0.4702 | 0.9998 |
| | MINI-ARC | 0.9953 | 0.5984 | 1.9939 | 0.4367 | 0.9737 |
| | List Fns | 0.9964 | 0.9542 | 3.3586 | 0.4962 | 0.9740 |
| | ARC-2025 | 0.9892 | 0.3456 | 2.0372 | 0.5536 | 0.8676 |
| Llama-3.3-70B-Instruct | Avg | 0.9660 | 0.2469 | 1.1862 | 0.2245 | 0.9645 |
| | ACRE | 0.9332 | 0.5930 | 1.1729 | 0.2008 | 0.9854 |
| | MINI-ARC | 0.9772 | 0.0569 | 1.1977 | 0.2352 | 1.0000 |
| | List Fns | 0.9695 | 0.3095 | 1.2279 | 0.1622 | 0.9858 |
| | ARC-2025 | 0.9842 | 0.0282 | 1.1460 | 0.2997 | 0.8867 |
| Qwen2.5-72B-Instruct | Avg | 0.9990 | 0.1848 | 1.3988 | 0.3996 | 0.9865 |
| | ACRE | 0.9987 | 0.4770 | 1.3119 | 0.3847 | 0.9998 |
| | MINI-ARC | 0.9983 | 0.0319 | 1.5144 | 0.5793 | 0.9784 |
| | List Fns | 1.0000 | 0.2086 | 1.4646 | 0.3511 | 0.9798 |
| | ARC-2025 | 0.9992 | 0.0215 | 1.3042 | 0.2835 | 0.9881 |
| gemma-2-9b-it | Avg | 0.9718 | 0.0522 | 1.0715 | 0.1857 | 0.9960 |
| | ACRE | 0.9893 | 0.1272 | 1.1661 | 0.4065 | 0.9999 |
| | MINI-ARC | 0.9892 | 0.0063 | 1.0000 | 0.0000 | 1.0000 |
| | List Fns | 0.9912 | 0.0729 | 1.1218 | 0.1507 | 0.9862 |
| | ARC-2025 | 0.9175 | 0.0025 | 0.9980 | – | 0.9980 |
| Qwen2.5-7B-Instruct | Avg | 0.9807 | 0.1036 | 1.1801 | 0.2714 | 0.9992 |
| | ACRE | 0.9597 | 0.3149 | 1.2475 | 0.3737 | 0.9971 |
| | MINI-ARC | 0.9885 | 0.0107 | 1.2061 | 0.3835 | 1.0000 |
| | List Fns | 0.9854 | 0.0861 | 1.2667 | 0.3285 | 0.9997 |
| | ARC-2025 | 0.9892 | 0.0029 | 0.9999 | 0.0000 | 0.9999 |
| Llama-3.1-8B-Instruct | Avg | 0.1505 | 0.0020 | 1.0735 | 0.5421 | 1.0000 |
| | ACRE | 0.0578 | 0.0048 | 1.1469 | 0.5421 | 1.0000 |
| | MINI-ARC | 0.2217 | 0.0000 | – | – | – |
| | List Fns | 0.1917 | 0.0031 | 1.0000 | – | 1.0000 |
| | ARC-2025 | 0.1308 | 0.0000 | – | – | – |
| NextCoder-7B | Avg | 0.9891 | 0.0704 | 1.1351 | 0.4410 | 0.9930 |
| | ACRE | 0.9911 | 0.2097 | 1.1562 | 0.3397 | 0.9999 |
| | MINI-ARC | 0.9842 | 0.0100 | 1.1550 | 0.5363 | 1.0000 |
| | List Fns | 0.9952 | 0.0612 | 1.2293 | 0.4470 | 0.9721 |
| | ARC-2025 | 0.9858 | 0.0006 | 1.0000 | – | 1.0000 |

Table 4: Ablation results: GEAR score & T3 accuracy under different training settings.

| GEAR *Scores* | | | | | |
|---|---|---|---|---|---|
| | Instruction Following Rate | Consistency | Generalizability | $\beta$ | $\gamma$ |
| Llama-3.1-8b | 0.149 | 0.002 | 1.000 | 0.000 | 1.000 |
| Llama-3.1-8b-dpo-fixed-ratio | 0.968 | 0.014 | 1.000 | 0.000 | 1.000 |
| Llama-3.1-8b-dpo-momentum-curriculum | **0.970** | **0.020** | 1.000 | **0.196** | **1.179** |
| Llama-3.1-8b-parsing-ablation | 0.941 | 0.009 | 1.000 | 0.000 | 1.000 |
| Llama-3.1-8b-consistent-ablation | 0.322 | 0.009 | 1.000 | 0.014 | 1.011 |
| Llama-3.1-8b-GEAR-ablation | 0.258 | 0.005 | 1.000 | 0.000 | 1.000 |
| qwen-2.5-7b | 0.988 | 0.037 | 0.987 | 0.032 | 1.001 |
| qwen-2.5-7b-dpo-fixed-ratio | 0.996 | 0.042 | 0.987 | 0.044 | 1.046 |
| qwen-2.5-7b-dpo-momentum-curriculum | **0.997** | 0.041 | 1.000 | **0.073** | **1.052** |
| qwen-2.5-7b-parsing-ablation | 0.996 | 0.032 | 1.000 | 0.014 | 1.008 |
| qwen-2.5-7b-consistent-ablation | 0.988 | **0.047** | 0.975 | 0.015 | 1.010 |
| qwen-2.5-7b-GEAR-ablation | 0.981 | 0.034 | 1.000 | 0.023 | 1.012 |
| nextcoder-7b | 0.863 | 0.013 | 1.000 | 0.140 | 1.081 |
| nextcoder-7b-dpo-fixed-ratio | 0.965 | 0.028 | 1.000 | 0.078 | 1.066 |
| nextcoder-7b-momentum-curriculum | 0.991 | 0.030 | 0.991 | 0.182 | **1.151** |
| nextcoder-7b-parsing-ablation | **0.992** | 0.020 | 1.000 | 0.222 | 1.140 |
| nextcoder-7b-consistent-ablation | 0.945 | **0.034** | 1.000 | 0.054 | 1.040 |
| nextcoder-7b-GEAR-ablation | 0.980 | 0.015 | 0.976 | **0.202** | 1.145 |

| T3 *accuracies* | | | | | |
|---|---|---|---|---|---|
| | Avg Train Pass Rate | Avg Test Pass Rate | Top-1 Acc. | Top-2 Acc. | Top-3 Acc. |
| Llama-3.1-8b | 0.011 | 0.009 | 0.000 | 0.003 | 0.004 |
| Llama-3.1-8b-dpo-fixed-ratio | 0.096 | 0.076 | 0.028 | 0.043 | 0.049 |
| Llama-3.1-8b-dpo-momentum-curriculum | 0.129 | **0.101** | **0.035** | **0.052** | **0.065** |
| Llama-3.1-8b-parsing-ablation | **0.130** | 0.099 | 0.026 | 0.031 | 0.041 |
| Llama-3.1-8b-consistent-ablation | 0.084 | 0.066 | 0.018 | 0.022 | 0.028 |
| Llama-3.1-8b-GEAR-ablation | 0.039 | 0.030 | 0.011 | 0.016 | 0.020 |
| qwen-2.5-7b | 0.198 | 0.154 | 0.035 | 0.059 | 0.068 |
| qwen-2.5-7b-dpo-fixed-ratio | 0.205 | 0.160 | 0.054 | 0.064 | 0.077 |
| qwen-2.5-7b-dpo-momentum-curriculum | 0.203 | 0.163 | **0.066** | **0.083** | **0.095** |
| qwen-2.5-7b-parsing-ablation | 0.198 | 0.154 | 0.045 | 0.055 | 0.066 |
| qwen-2.5-7b-consistent-ablation | **0.224** | **0.176** | 0.059 | 0.075 | 0.086 |
| qwen-2.5-7b-GEAR-ablation | 0.183 | 0.141 | 0.039 | 0.056 | 0.065 |
| nextcoder-7b | 0.151 | 0.121 | 0.025 | 0.044 | 0.055 |
| nextcoder-7b-dpo-fixed-ratio | 0.173 | 0.136 | 0.051 | 0.065 | 0.074 |
| nextcoder-7b-momentum-curriculum | 0.189 | 0.149 | **0.065** | **0.080** | **0.092** |
| nextcoder-7b-parsing-ablation | 0.162 | 0.125 | 0.045 | 0.055 | 0.062 |
| nextcoder-7b-consistent-ablation | **0.200** | **0.153** | 0.060 | 0.074 | 0.080 |
| nextcoder-7b-GEAR-ablation | 0.150 | 0.118 | 0.033 | 0.043 | 0.049 |

# D  FIGURES

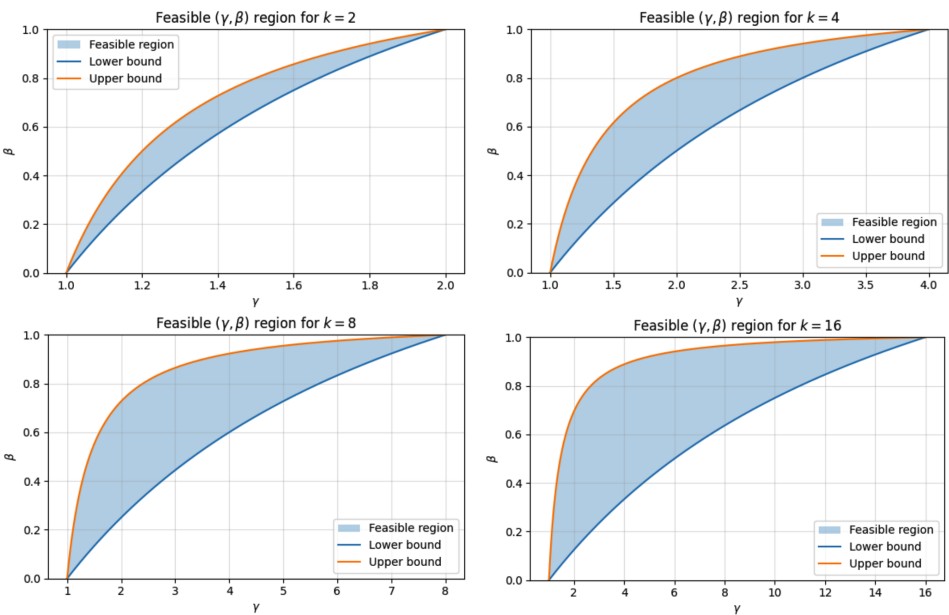

Figure 5: Feasible region for $(\gamma, \beta)$

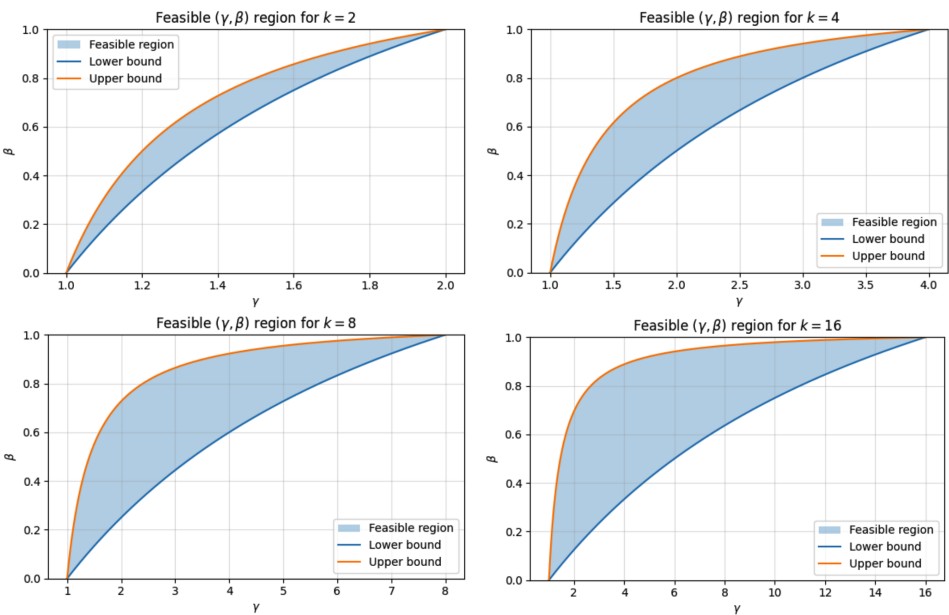

Figure 6: Training log on single-preference DPO

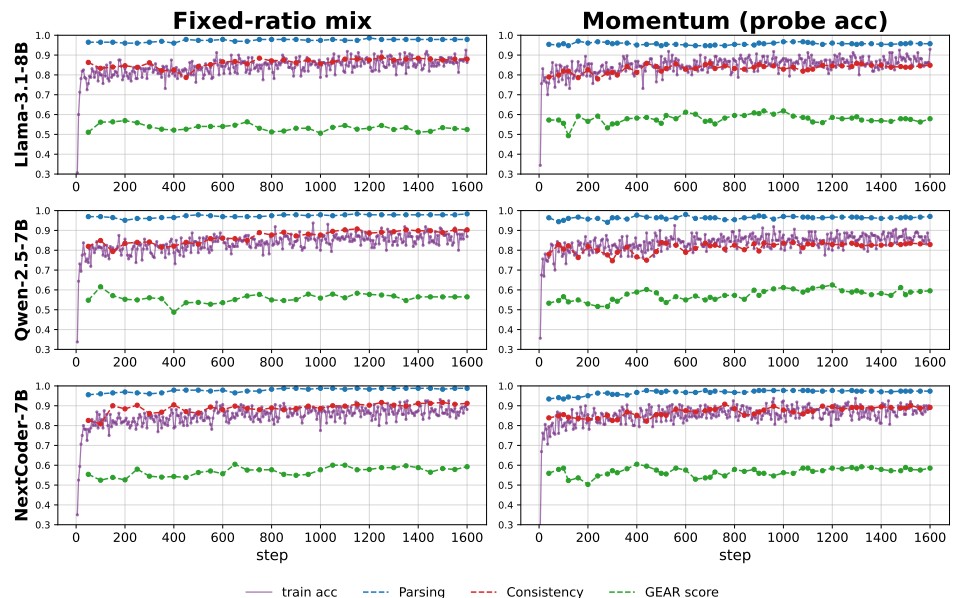

Figure 7: Training log on multi-preference DPO

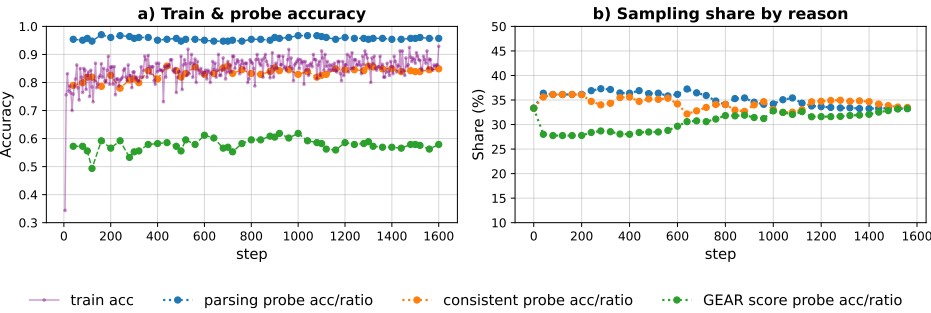

Figure 8: Momentum curriculum training log for Llama-3.1-8B

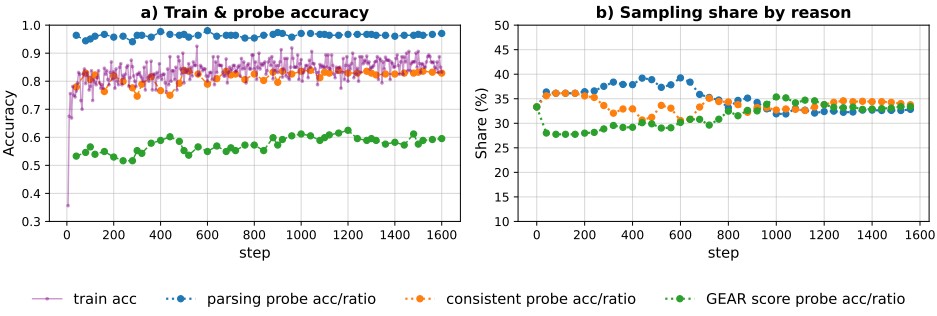

Figure 9: Momentum curriculum training log for Qwen-2.5-7B

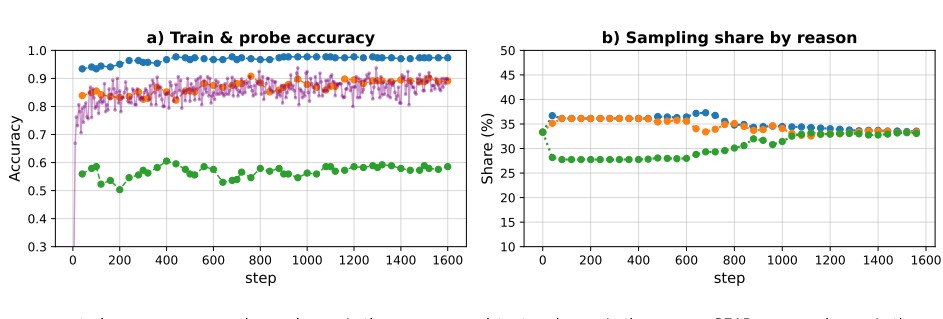

Figure 10: Momentum curriculum training log for NextCoder-7B

## E  TRAINING LOG ANALYSIS

Across our training signals, the three preferences exhibit a clear *hierarchy*: a hypothesis must first be parsable to be eligible for consistency checking; only hypotheses that pass consistency check can contribute to generalizability and diversity. This structure implies that optimizing for a single preference in isolation is insufficient. As shown in Figure 6, training on only one preference can raise the corresponding validation accuracy, but typically fails to improve—and often degrades—the other two. For example, parsing-only training does not translate into higher consistency or GEAR gains. Consistent with Table 4, parsing-only runs also do not yield a meaningful increase in the diversity of generated hypotheses nor in downstream T-3 accuracy. Taken together, these results indicate that leveraging all three preferences during training is necessary to realize balanced improvements across GEAR.

Figure 7 further compares fixed-ratio mixing with our *momentum curriculum*. Although momentum-trained models show slightly lower peak accuracy on parsing and consistency early on, they make steadier progress on the GEAR preference and reach roughly 60% validation accuracy during training. This translates into the strongest diversity scores and the best downstream T-3 accuracy in subsequent evaluations (see Table 4). The curriculum's adaptive reweighting prioritizes what is learning fastest while gradually shifting emphasis to harder signals, yielding a more stable training trajectory and superior overall abduction quality.

## F  MATHEMATICAL RELATIONSHIP BETWEEN $\beta$ & $\gamma$

**Setup.**  Let $\mathbb{S}$ be the (finite) sample space of inputs with $|\mathbb{S}| = n$, and let $\mathbb{F} = \{f_1, \ldots, f_k\}$ be $k$ hypotheses. For each $f \in \mathbb{F}$, recall the *prediction set* on $\mathbb{S}$:

$$\mathbb{P}_f \; := \; \{ \, (\mathrm{in}, f(\mathrm{in})) \; : \; \mathrm{in} \in \mathbb{S} \, \}$$

We work under the equal-size assumption (standard in our experiments): every $f \in \mathbb{F}$ produces exactly one prediction per input in $\mathbb{S}$, hence $|\mathbb{P}_f| = n$.[1] Define the union size

$$U \; := \; \Big| \bigcup_{f \in \mathbb{F}} \mathbb{P}_f \Big|, \qquad \gamma(\mathbb{F}; \mathbb{S}) \; = \; \frac{U}{|\mathbb{S}|} \; = \; \frac{U}{n} \in [1, k],$$

so $\gamma$ is the average number of unique predictions per input.

**Multiplicity $c_x$ and the identity $\sum_x c_x = kn$.**  For each element $x$ in the union $\bigcup_{f \in \mathbb{F}} \mathbb{P}_f$, define its *reuse multiplicity*

$$c_x \; := \; \#\{ \, f \in \mathbb{F} : x \in \mathbb{P}_f \, \} \in \{1, \ldots, k\}.$$

Intuitively, $c_x$ counts in how many hypotheses' prediction sets the *same prediction pair* $x = (\mathrm{in}, \mathrm{out})$ appears. By *double counting*,

$$\sum_x c_x \; = \; \sum_x \sum_{f \in \mathbb{F}} \mathbf{1}\{x \in \mathbb{P}_f\} \; = \; \sum_{f \in \mathbb{F}} \sum_x \mathbf{1}\{x \in \mathbb{P}_f\} \; = \; \sum_{f \in \mathbb{F}} |\mathbb{P}_f| \; = \; kn.$$

The quantity

$$r \; := \; \frac{kn}{U} \; = \; \frac{k}{\gamma} \; \in [1, k]$$

is the *average reuse multiplicity*: on average, each distinct prediction pair in the union is reused by $r$ hypotheses.

**Pairwise intersections and the definition of $t$.**  For a pair $(f_i, f_j)$ with $i < j$, define

$$t_{ij} \; := \; \big| \mathbb{P}_{f_i} \cap \mathbb{P}_{f_j} \big| \; \in \; [0, n],$$

namely, the number of inputs in $\mathbb{S}$ on which $f_i$ and $f_j$ make the *same* prediction pair $\big(\mathrm{in}, f(\mathrm{in})\big)$. Let the total and the average pairwise intersection sizes be

$$I \; := \; \sum_{1 \le i < j \le k} t_{ij} \; = \; \sum_x \binom{c_x}{2} \; = \; \frac{1}{2}\Big( \sum_x c_x^2 - kn \Big), \qquad \bar{t} \; := \; \frac{I}{\binom{k}{2}}.$$

---

[1]If a hypothesis can be undefined on some inputs, one may attach a sentinel output $\perp$; this preserves $|\mathbb{P}_f| = n$ without changing the extremal structure below.

**Jaccard similarity** $\text{sim}(t)$ **and why** $t/(2n - t)$**.** For two finite sets $A, B$, the Jaccard *similarity* is

$$\text{Jacc}(A, B) \;=\; \frac{|A \cap B|}{|A \cup B|}.$$

Here $A = \mathbb{P}_{f_i}$ and $B = \mathbb{P}_{f_j}$ satisfy $|A| = |B| = n$ by assumption, so if we denote $t = |A \cap B| = t_{ij}$, then $|A \cup B| = |A| + |B| - |A \cap B| = 2n - t$, and

$$\text{sim}(t) \;:=\; \text{Jacc}(\mathbb{P}_{f_i}, \mathbb{P}_{f_j}) \;=\; \frac{t}{2n - t}, \qquad t \in [0, n].$$

Our $\beta$-diversity is the average Jaccard *dissimilarity* across pairs, i.e.,

$$\beta(\mathbb{F}; \mathbb{S}) \;=\; \frac{2}{k(k-1)} \sum_{i<j} \Big(1 - \text{sim}(t_{ij})\Big), \quad \text{so} \quad 1 - \beta \;=\; \frac{2}{k(k-1)} \sum_{i<j} \text{sim}(t_{ij}).$$

**Step 1: Feasible range of the average intersection.** By Cauchy–Schwarz,

$$\sum_x c_x^2 \;\geq\; \frac{(\sum_x c_x)^2}{U} \;=\; \frac{(kn)^2}{U} \;=\; kn\,r,$$

hence

$$\boxed{\bar{t}_{\min} \;=\; \frac{n(r-1)}{k-1}}.$$

For a matching upper envelope, concentrate as many $c_x$'s at $k$ as possible and set the rest to $1$. If $t$ elements have $c_x = k$ and all others have $c_x = 1$, the constraint $\sum_x c_x = kn$ gives $t = \dfrac{kn - U}{k-1} = \dfrac{kn(r-1)}{r(k-1)}$, and every pair shares exactly these $t$ elements, so

$$\boxed{\bar{t}_{\max} \;=\; \frac{kn(r-1)}{r(k-1)}}.$$

Clearly $\bar{t}_{\min} \leq \bar{t}_{\max}$ with equality only at $r \in \{1, k\}$.

**Step 2: Bounds for the average similarity** $1 - \beta$**.** The map $\text{sim}(t) = \dfrac{t}{2n - t}$ is increasing and convex on $[0, n]$. By Jensen,

$$1 - \beta \;=\; \frac{2}{k(k-1)} \sum_{i<j} \text{sim}(t_{ij}) \;\geq\; \text{sim}(\bar{t}) \;\geq\; \text{sim}(\bar{t}_{\min}),$$

and the "$k$-shared-core + disjoint-uniques" construction achieves the upper envelope $1 - \beta = \text{sim}(\bar{t}_{\max})$. Therefore,

$$\boxed{\frac{r-1}{2(k-1) - (r-1)} \;\leq\; 1 - \beta \;\leq\; \frac{k(r-1)}{k(r+1) - 2r}}, \qquad r = \frac{k}{\gamma}.$$

**Step 3: Bounds expressed purely via** $\gamma$**.** Substituting $r = \frac{k}{\gamma}$ and simplifying yields closed forms:

$$\boxed{\frac{k - \gamma}{2k\gamma - \gamma - k} \;\leq\; 1 - \beta \;\leq\; \frac{k - \gamma}{k + \gamma - 2}}, \qquad \gamma \in [1, k].$$

Equivalently, for the dissimilarity itself,

$$\boxed{\frac{2(\gamma - 1)}{k + \gamma - 2} \;\leq\; \beta \;\leq\; \frac{2k(\gamma - 1)}{2k\gamma - \gamma - k}}, \qquad \gamma \in [1, k].$$

See Figure 5 in Appendix D for a visualization of the feasible region for different values of $k$.

**Edge cases.** At $\gamma = 1$ (all hypotheses make identical predictions on every input), both bounds give $\beta = 0$. At $\gamma = k$ (all predictions are pairwise disjoint on every input), both bounds give $\beta = 1$. Thus the bounds are tight at the extremes.

Dropping the equal-size assumption $|\mathbb{P}_{f_i}| = n$ requires replacing $\mathrm{sim}(t)$ by the general Jaccard formula $t/(n_i + n_j - t)$ and tracking per-pair sizes $(n_i, n_j)$. The same extremal principles still apply; small integrality effects only perturb the finite-sample envelopes by $O(1/n)$.

## G  PROMPTS

---

**Initialization prompt $P_{\mathrm{init}}$**

```
You must return one tuple of two raw strings (no Markdown fences, no back-ticks).

   element 0 = concise natural-language hypothesis
   element 1 = FULL Python source of exactly one top-level "def"

Code rules (apply to the source string in element 1)
- built-ins only (do not import anything)
- spaces-only indentation (4 spaces), "\n" newlines (no "\r")
- every control-flow header (if/for/while/else/elif/with/try) must break onto
  the next line; never place another statement after a colon
- at most 80 characters per line
- the file must compile with ast.parse() and execute with exec() unchanged
- do not add prints, tests, or extra defs; a return must appear in the function
- logic must generalize beyond the given pairs; no hard-coding

Task
- Below are (input, output) pairs \O_n .
- Infer one rule consistent with all pairs and write a function that follows it
  on unseen inputs.

Pairs: {{OBS_PAIRS}}

Return only:

(
 "My hypothesis in one sentence ...",
 "def f(x):\n    # your code\n    return y"
)

Example format (strictly follow):

(
 "Return 6 if 6 appears, else 0","
def f(x):\n
    if 6 in x:\n
        return [6]\n
    return [0]"
)

Note: This template is dataset-agnostic; only \O_n is instantiated per task.
```

---

---

**Iterative prompt $P_{\text{iter}}$**

```
Return one tuple of two raw strings (no Markdown fences, no back-ticks).

  element 0 = concise description of a new hypothesis
  element 1 = FULL Python source of exactly one top-level "def"

Code rules (identical to P_init)
- built-ins only; spaces-only indentation (4); "\n" newlines
- control-flow headers must break onto the next line; no statements after colon
- <= 80 chars per line; must compile with ast.parse() and run with exec()
- no prints/tests/extra defs; the function must contain a return
- generalize beyond the pairs; no hard-coding

You have proposed the following hypotheses so far:
F_{t-1} = {f_1, ..., f_{t-1}} (summaries below)
{{PREVIOUS_HYPOTHESES}}   # e.g., bullet list of brief natural-language hypotheses

Re-examine the (input, output) pairs \O_n :
{{OBS_PAIRS}}

Your goal
- Invent a brand-new hypothesis f_t that
  (i) is consistent with all pairs in \O_n, and
  (ii) is distinct in underlying principle from every f in F_{t-1}.

Return exactly:

(
 "Concise description of the new hypothesis",
 "def f(x):\n    # your code\n    return y"
)

Example format (strictly follow):

(
 "Return 6 if 6 appears, else 0"," 
def f(x):\n
    if 6 in x:\n
        return [6]\n
    return [0]"
)

Note: This template is shared across datasets; only \O_n and F_{t-1} vary.
```

---

# H USE OF LARGE LANGUAGE MODELS

We used large language models solely for light language editing—polishing phrasing and detecting grammar/spelling/typos. No ideas, analyses, algorithms, datasets, annotations, or paper results were generated or modified by LLMs. All technical content was authored and verified by the authors, and no non-public or personal data were shared with LLMs.

