# OpenReview forum: "GEAR: A $\textbf{G}$eneral $\textbf{E}$valuation Framework for $\textbf{A}$bductive $\textbf{R}$easoning"
_ICLR.cc/2026/Conference — Submitted to ICLR 2026_

### Official Review · Reviewer_brvY · 2025-10-18

**Soundness:** 2
**Presentation:** 3
**Contribution:** 2
**Rating:** 4
**Confidence:** 3

**Summary:**

This paper introduces GEAR, an automated, label-free evaluation framework intended to assess abductive reasoning capabilities in large language models. Empirical results across nine models show that GEAR reveals finer-grained performance differences than traditional gold-label metrics  and that the momentum curriculum improves GEAR scores and modestly enhances downstream abductive benchmarks.

**Strengths:**

- The paper convincingly argues that existing abductive reasoning benchmarks relying on single "gold" labels or human judgments are inadequate for underdetermined reasoning problems.
- The paper evaluates 9 diverse LLMs across 4 datasets and provides simulation studies demonstrating that abductive reasoning is defeasible

**Weaknesses:**

- Although the framework is titled General Evaluation for Abductive Reasoning, all experiments are performed on programmable and synthetic tasks (e.g., ARC, ACRE, LIST FUNCTIONS).

- The β/γ-diversity measures are based purely on output pattern dissimilarity (set overlap/Jaccard distance) - may capture syntactic or behavioral heterogeneity, not genuine content diversity.

- While the proposed adaptive weighting method is sensible, the results (Table 2, 4) show only marginal gains, and the analysis of why or how these improvements occur is minimal. The ablation studies offer little evidence of actual reasoning enhancement.

**Questions:**

- The β/γ-diversity metrics are based on prediction overlap. How do you ensure these metrics capture conceptual rather than purely behavioral diversity?

- Have you analyzed qualitative examples of generated hypotheses to verify that improvements in β/γ-diversity correspond to more meaningful explanations?

---

> ### Author Response · Authors · 2025-11-15
> **Response to reviewer brvY - Post (1)**
>
> Thank you for your thoughtful comments and suggestions. We would like to address your concerns as follows:
>
> ### **Response to Weakness 1:**
> - **Why choose programmable tasks instead of NL tasks**: Research on abductive reasoning with LLMs is still in its early stages, having recently shifted from formal logic-based tasks (like first-order logic) to more complex tasks based on programming languages. Based on our literature review, most work on abductive reasoning relies on synthetic tasks because when Natural Language (NL) is introduced, it becomes difficult to perform convincing evaluations. Tasks beyond synthetic ones are often evaluated by humans or an LLM-as-judge, but these evaluations rely on subjective metrics like surprisingness, novelty, and excitingness [8], which makes benchmarks based on such evaluations subjective, non-reproducible, and expensive [1],[2],[7]. For instance, if you propose a new method for these benchmarks, you must hire human evaluators, yet you cannot hire the same batch of people for every evaluation to get consistent results, and these high-level metrics cannot provide the direct feedback needed to update and optimize your method. This phenomenon is rooted in the lack of a systematic understanding of reasoning; beyond LLMs themselves, we lack effective tools for NL-level reasoning. However, code and executable programs represent a middle ground, offering flexibility and expressiveness similar to NL while still permitting verifiable deductive evaluation. Until we have fully explainable and convincing deductive systems for NL, the field of abductive reasoning will likely remain focused on synthetic tasks. (We note that ARC is an up-to-date and challenging dataset, even for modern LLMs).
>
> - **Why choose synthetic tasks over real tasks**: Since abductive reasoning tests the ability of LLMs to learn **NEW** knowledge, the very basic requirement is that LLMs do not know such knowledge beforehand. Synthetic tasks are the main choice in current studies for two key benefits: 1) The tasks are designed from scratch and are unlikely to have been seen during pretraining, ensuring LLMs are tested on learning new knowledge rather than recalling memorized facts (e.g., benchmark questions in papers [1] and [2] can be answered from pretrained memory). 2) The evaluation is convincing and can be reused in other research. Given the tools we currently have, synthetic tasks remain the optimal choice for evaluating the abductive reasoning ability of LLMs.
>
> - Our contribution is to identify the logical problem in existing evaluations that assume a single gold hypothesis is the only correct answer. We shift the evaluation from a single hypothesis to a set of hypotheses, focusing on the expressive power of the set rather than a single output hypothesis. This represents an evaluation paradigm shift.  A qualitative example (see response to Question 1) shows that our GEAR signal can not only improve the consistency of generated hypotheses but also provide more novel and creative perspectives.
>
> ### **Response to Weakness 2:**
> - In fact, we argue the opposite: for hypothesis correctness, we should care more about the prediction pattern (i.e., functional equivalence) than the superficial content (i.e., syntactic/textural form).
> - Some existing benchmarks (e.g., paper [1]) directly compare the content of a generated hypothesis to a labeled one using token-level matching. For example, a labeled hypothesis might be, “If a plant is carnivorous, then it probably has a trapping structure.” A model might generate, “A carnivorous plant is able to trap insects.” While a human would recognize these as semantically similar, token-matching metrics like ROUGE or METEOR and embedding method like BERTScore would yield very different results. To make NL tasks evaluable under this paradigm, researchers must constrain hypotheses to rigid "if-else" or other structured statements, which are sometimes even less expressive than programming languages.
> - Due to the polysemy of NL, it is unreliable to evaluate the true "content" of a hypothesis. This is also true in programming languages if we directly evaluate its “content”. The most popular ways currently to evaluate code similarity are CodeBERTScore (which does not solve the polysemy problem) or AST parsing (which cannot equate semantics like a recursive function and an iterative loop that are functionally identical). Therefore, given the choices of Textual Similarity (CodeBERTScore), Structural Similarity (AST parsing), and Functional Equivalence (evaluation on test cases), most researchers choose Functional Equivalence (i.e., the prediction pattern) as the primary evaluation method.
> - Continued in the second post.

---

> ### Author Response · Authors · 2025-11-15
> **Response to reviewer brvY - Post (2)**
>
> The reason most benchmarks use a train-test split is based on this assumption: if the hypothesis content is correct, it will produce the correct prediction pattern. This means current practice, as demonstrated by most benchmarks, already uses prediction patterns as a more reliable representative for content semantics. As we can see, most popular benchmarks for abductive or inductive reasoning are either highly structured to eliminate polysemy  (e.g., [5],[6] first-order logic) or they use prediction patterns ([9],[10]) to evaluate correctness.
>
> ### **Response to weakness 3:**
> - The improvement in **instruction following** was not large for Qwen and NextCoder, as their base rates were already high (>85%). However, our method proved highly effective for Llama-3, boosting its performance from 14.9% to over 95%. This demonstrates our training method's effectiveness in enhancing instruction-following capabilities.
> - Regarding generalizability, we found that current LLMs already perform well, proposing hypotheses that generalize beyond the given training cases instead of just overfitting. As a result, the base generalizability scores are already high, leaving minimal room for improvement.
> - For diversity, the improvement may appear small at a glance, but this is because the metric's absolute scale is small. As shown by the feasible region in Figure 5 (Appendix D) and proven in Appendix F, given a set of $k$ hypotheses, the acceptable range for $\beta$ is narrow. For example (Qwen-2.5-7B), for $k=3$ hypotheses (the number we generate at test time), if the average $\gamma = 1.046$, the entire possible range for $\beta$ is only `[0.04, 0.12]`. If average $\gamma = 1.052$, the range is `[0.05, 0.14]`.
> - When scaled to this feasible range, the improvement is not marginal. For Qwen-2.5-7B, the improvement from fixed-ratio DPO ($\beta=0.044$) to our momentum curriculum ($\beta=0.073$) (Table 2) represents a significant jump within that tiny window. When normalized, this is a **substantial gain (approx. 0.05 to 0.26 on a normalized 0-1 scale)**, not a marginal one. The improvements for Llama-3.1 and NextCoder-7B are even larger.
> - As for $\gamma$ diversity, this is a very difficult metric to improve. As shown in Table 3, a model 10x larger (Llama-3.3-70B) only achieves $\gamma \approx 1.18$ (even when not limited to 3 hypotheses). Yet after curriculum DPO, next-coder 7B and Llama-8B have on average 0.1 more unique predictions generated per input, with a sample size of 1000 that would be 100 unique predictions. Therefore, the increases we achieved for NextCoder-7B and Llama-3.1-8B should not be considered minimal.
> - We provide a comprehensive ablation study in Table 4. To analyze the effect of our three preference types (instruction following/parsing, consistency, and GEAR score), we compare DPO training across several settings: using a single preference source, using a mixed preference source, and using our proposed dynamic curriculum with a mixed preference source. The results show that our proposed curriculum DPO has the best performance across all settings. This is sufficient to demonstrate that only training with a mixed preference source and a dynamic curriculum can achieve the best performance, which highlights the effectiveness of our method.
> - Training with DPO improves performance because the GEAR score provides a sufficient signal to guide the LLM. Based on a given observation set $\mathbb{O}_{init}$ and existing hypotheses $H$, the score indicates which new hypotheses will add more diverse and genuinely new prediction patterns. Consequently, the LLM learns to provide novel perspectives rather than just making minor refinements to previous hypotheses. The quantitative results of this enhancement can be seen in Tables 2 and 4, while a qualitative example is provided in our response to Question 1.

---

> ### Author Response · Authors · 2025-11-15
> **Response to reviewer brvY - Post (3)**
>
> ### **Response to Questions 1 & 2:**
> - Please see our detailed response to Weakness 2. As stated there, content-wise metrics (such as Textual or Structural Similarity) are unable to reliably define the essence of a hypothesis. Furthermore, "meaningful explanations" is not a well-defined metric and relies heavily on subjective human judgment. The current best practice to capture the core of a hypothesis is to analyze its prediction pattern. Indeed, classic theories like the 'multiple working hypothesis' [3] and 'strong inference' [4] are also built upon this principle: i.e., in real scientific discovery, we care whether two hypotheses are able to produce different predictions on an input, so that we can design experiments to verify and distinguish between these hypotheses. The prediction pattern is the hypothesis's true essence; the textual representation is simply the medium used to communicate that pattern.
> - However, as shown in our prompt (Appendix G), we do require the LLM to generate an NL description alongside its code hypothesis. We performed a small-scale quality check on ~1,000 (Qwen-2.5-7B) samples before the main experiment and confirmed that the generated NL descriptions were well-aligned with their corresponding code logic. Furthermore, to prevent trivial solutions, we implemented an 'AST-based cheating check' to automatically flag hypotheses that are merely lookup tables (e.g., `if x == a: return b...`). We confirm that none of the 50,340 hypotheses in our final dataset are simple lookup tables; all are genuine, compressed hypotheses attempting to find a general rule for the I/O pairs. We provide an example set below, randomly sampled from a lower-performing model (Nextcoder-7B), and will release all generated hypotheses upon acceptance.
>
>   - **Question Prompt: (From ACRE dataset question id 66)**
>     - " Below are (input, output) pairs. Infer the rule and write one function that works for unseen inputs following the same rule. Pairs: Pair0: (['blue rubber cylinder', 'green rubber cube'], on) Pair1: (['blue rubber cylinder'], on) Pair2: (['green rubber cube'], off) Pair3: (['gray rubber cylinder'], off) Pair4: (['cyan rubber cube'], on) Pair5: (['brown metal sphere', 'red rubber sphere'], off) Return **only**: ( "My hypothesis in one sentence …", "def f(x):\n # your code\n return y" ) Example format (strictly follow): ( "Return 6 if 6 appears, else 0"," def f(x):\n if 6 in x:\n return [6]\n return [0]" ) "
>   - Generated Hypotheses (**BEFORE** Training):
>     - **H1:** ("Turn on if there is a blue rubber cylinder, otherwise turn off", `def f(x):\n if 'blue rubber cylinder' in x:\n return 'on'\n return 'off'`)
>     - **H2:** ("Turn on if the count of 'blue rubber cylinder' is odd, otherwise turn off", `def f(x):\n count = x.count('blue rubber cylinder')\n return 'on' if count % 2 == 1 else 'off'`)
>     - **H3:** ("Turn on if any object is 'blue rubber cylinder', otherwise turn off", `def f(x):\n return 'on' if 'blue rubber cylinder' in x else 'off'`)
>   - Generated Hypotheses (**AFTER** Training):
>     - **G1:** ("Return 'on' if the list contains 'blue rubber cylinder' or 'cyan rubber cube', else 'off'", `def f(x):\n if 'blue rubber cylinder' in x or 'cyan rubber cube' in x:\n return 'on'\n return 'off'`)
>     - **G2:** ("Return 'on' if the number of items in the list is odd, else 'off'", `def f(x):\n if len(x) % 2 == 1:\n return 'on'\n return 'off'`)
>     - **G3:** ("Return 'on' if any color starts with 'b', 'c', or 'g' (irrespective of material), else 'off'", `def f(x):\n colors = [item.split()[0] for item in x]\n if any(color in ['blue', 'brown', 'cyan', 'green'] for color in colors):\n return 'on'\n return 'off'`)
>   - As you can see, the Before Training set (H1-H3) is not very diverse; all three hypotheses are minor variations focused only on the single 'blue rubber cylinder' feature. H1, H2, and H3 are largely functionally identical. The After Training set (G1-G3) is far more conceptually diverse. G2 explores a meta-feature (the list's length, `len(x)`), and G3 explores a completely novel explanatory angle (abstract color properties, `item.split()[0]`) that the base model did not find.
>   - **Most importantly, this increase in novelty was accompanied by a dramatic improvement in consistency.** The Before Training hypotheses (H1-H3), being largely functionally identical, are all inconsistent with the full set of observations. In contrast, the After Training set, by exploring new perspectives, successfully finds a fully consistent hypothesis (G1) that correctly explains all 6 cases. This shows the LLM learned to find a correct rule, not just a different one. This dual improvement is reflected in the scores for this single problem:
>   - **Consistency (Avg. Cases Matched):**
>     - Before: 0 / 3 hypotheses are consistent.
>     - After: 1 / 3 hypotheses is fully consistent (G1).
>   - **Diversity (Total Unique Mappings):**
>     - Before: 7,049
>     - After: 13,206

---

> ### Author Response · Authors · 2025-11-15
> **References**
>
> - [1] Yang, Zonglin, et al. "Language models as inductive reasoners." Proceedings of the 18th Conference of the European Chapter of the Association for Computational Linguistics (Volume 1: Long Papers). 2024.
> - [2] Wang R, Jansen P, Côté MA, Ammanabrolu P. Scienceworld: Is your agent smarter than a 5th grader?. arXiv preprint arXiv:2203.07540. 2022 Mar 14.
> - [3] Thomas C Chamberlin. The method of multiple working hypotheses: With this method the dangers of parental affection for a favorite theory can be circumvented. Science, 148(3671):754–759,1965.
> - [4]John R Platt. Strong inference: Certain systematic methods of scientific thinking may produce much more rapid progress than others. science, 146(3642):347–353, 1964.
> - [5] Young, Nathan, et al. "Abductionrules: Training transformers to explain unexpected inputs." Findings of the Association for Computational Linguistics: ACL 2022. 2022.
> - [6] He J, Lu F. Causejudger: Identifying the cause with llms for abductive logical reasoning. arXiv preprint arXiv:2409.05559. 2024 Sep 9.
> - [7] Jansen, Peter, et al. "Discoveryworld: A virtual environment for developing and evaluating automated scientific discovery agents." Advances in Neural Information Processing Systems37 (2024): 10088-10116.
> - [8] Si C, Yang D, Hashimoto T. Can llms generate novel research ideas? a large-scale human study with 100+ nlp researchers. arXiv preprint arXiv:2409.04109. 2024 Sep 6.
> - [9] Li J, Cao P, Jin Z, Chen Y, Liu K, Zhao J. Mirage: Evaluating and explaining inductive reasoning process in language models. arXiv preprint arXiv:2410.09542. 2024 Oct 12. (ICLR 2025)
> - [10] Liu, Emmy, Graham Neubig, and Jacob Andreas. "An incomplete loop: Instruction inference, instruction following, and in-context learning in language models." arXiv preprint arXiv:2404.03028 (2024). (COLM 2024)

---

> ### Author Response · Authors · 2025-11-19
> **Follow-up on Rebuttal**
>
> Dear Reviewer bray,
>
> Hope you're having a productive week.
>
> We're following up on our rebuttal, which we posted four days ago. We hope the clarifications provided successfully addressed your concerns. We would be grateful if you could kindly provide any brief feedback or confirmation regarding whether the main issues are now resolved.
>
> Thanks for your time and guidance on our paper!
>
> Best, The Authors

---

> ### Comment · Reviewer_brvY · 2025-11-24
> **Response to the Authors**
>
> Reviewer Response:
>
> Thank you for the rebuttal and follow-up clarification. However, while the responses help clarify certain design choices, they do not fully resolve my core concerns regarding the generality of the framework and whether the diversity metrics truly capture conceptual variation. Given these remaining uncertainties, I will keep my current evaluation and score unchanged.

---

> ### Author Response · Authors · 2025-11-26
> **Response to Reviewer brvY - Post (4)**
>
> Dear Reviewer brvY,
> We understand your primary remaining concern is whether our label-free diversity metrics ( $\beta$ / $\gamma$ ), which are based on prediction patterns, truly correspond to **conceptual diversity** (i.e., distinct reasoning logic) rather than merely syntactic or behavioral noise.
> To empirically resolve this uncertainty, we conducted two new layers of evaluation: an **LLM-as-a-Judge** study (using GPT-5-Nano) and a **blind Human Evaluation**.
> ### **Experimental Setup**
> - A. LLM-as-a-Judge: We instructed GPT-5-Nano to act as a Senior Computer Scientist evaluating "Algorithmic Diversity." The model was prompted to ignore variable names and syntactic sugar, scoring solutions (0-10) based on whether they explore distinctly different logical perspectives.
> - B. Human Evaluation: To validate the automated judge, we hired 4 evaluators:
>   - 2 External CS graduate students (unrelated to this paper).
>   - 2 Co-authors.
>   - Protocol: We developed a blind evaluation script. Evaluators were shown the problem description and the generated hypotheses but were *blinded* to the model source (Base vs. DPO vs. Momentum). Human evaluators received the exact same instructions and prompts as the LLM judge.
>   - Sampling: Due to the time-intensive nature of human review, we uniformly subsampled 200 distinct problems.
>
> ### **Results & Analysis**
> The results, presented in the table below, strongly support that GEAR metrics drive genuine conceptual diversity.
>
> | Model | Setting | SBERT Cos | 1-SelfBLEU | CodeBERT | LLM Judge | Human1* (Ext) | Human2 (Co) | Human3 (Co) | Human4* (Ext) |
> | :--- | :--- | :--- | :--- | :--- | :--- | :--- | :--- | :--- | :--- |
> | **Qwen2.5-7B** | Base | 0.0465 | 0.2069 | 0.0005 | 4.03 | 1.53 | 2.50 | 2.71 | 2.12 |
> | **Qwen2.5-7B** | DPO Fix Ratio | 0.0542 | 0.2282 | 0.0008 | 4.22 | 1.67 | 2.44 | 2.35 | 1.95 |
> | **Qwen2.5-7B** | **Momentum** | 0.0417 | 0.1881 | 0.0004 | **4.34** | **1.86** | **2.97** | **3.19** | **2.17** |
> | **Llama-3.1-8B** | Base | 0.0457 | 0.1806 | 0.0008 | 2.62 | 0.65 | 1.15 | 1.25 | 1.45 |
> | **Llama-3.1-8B** | DPO Fix Ratio | 0.0929 | 0.2257 | 0.0003 | **4.18** | 0.94 | 1.81 | 1.32 | 1.56 |
> | **Llama-3.1-8B** | **Momentum** | 0.0728 | 0.1863 | 0.0004 | 3.37 | **1.40** | **2.67** | **1.87** | **1.73** |
> | **NextCoder-7B** | Base | 0.0219 | 0.0590 | 0.0003 | 3.62 | 1.29 | 2.00 | 1.57 | 1.07 |
> | **NextCoder-7B** | DPO Fix Ratio | 0.0680 | 0.1634 | 0.0006 | 3.93 | 1.04 | 2.15 | 1.81 | 1.69 |
> | **NextCoder-7B** | **Momentum** | 0.0408 | 0.1258 | 0.0005 | **4.15** | **1.62** | **3.57** | **3.33** | **2.29** |
>
> *\*Human1 and Human4 are external evaluators.*
>
> **Key Insights:**
> - **1. Prediction-Based Optimization Creates Conceptual Diversity**
>   You asked how we ensure prediction metrics ( $\beta$ / $\gamma$ ) capture conceptual diversity. The data provides the empirical link:
>   - Our Momentum Curriculum relies *solely* on prediction patterns as the reward signal (no semantic supervision). Yet, it achieves the **highest Conceptual Diversity scores** across almost all human and LLM evaluators.
>   - For **Qwen2.5-7B**, Human 1 scores improved from 1.53 (Base) to **1.86 (Momentum)**, and Human 2 improved from 2.50 to **2.97**.
>   - For **NextCoder-7B**, the improvement is even more drastic. Human 3 ratings jumped from 1.57 (Base) to **3.33 (Momentum)**.
>   - **Conclusion:** A model cannot produce diverse, consistent prediction patterns on unseen data merely by varying syntax; it *must* alter its underlying logic. Therefore, optimizing $\beta$ / $\gamma$ forces the model to learn conceptual diversity.
>
> - **2. Human-LLM Alignment & Qualitative Analysis**
>   We observed high consistency between the LLM Judge and Human Evaluators. Both groups agree that the Momentum Curriculum yields the most logically diverse solutions. This validates that our automated pipeline aligns with human perception of "novelty."
>   - **Qualitative Example (ACRE Dataset):**
>     - **Pre-training (Base Model):** Generated hypotheses often fail to extract diverse features, defaulting to rigid entity matching. The most widely adopted pattern typically uses the entire entity description for a hard check (e.g., `if 'blue rubber cylinder' in x: return 'on'`), lacking abstraction.
>     - **Post-training (Momentum):** Hypotheses begin to extract features in novel ways. For example, instead of checking for specific objects, the model derives abstract rules such as: *Check if 'red metal cube' is the first or last item in the list*, `def f(x):\n if x and (x[0] == 'red metal cube' or x[-1] == 'red metal cube'):\n return 'on'\n return 'off'`.
>     - **Analysis:** The shift from surface-level entity matching to abstract property reasoning (like parity, counts, order, and other meta-features) represents a fundamental change in perspective. **This distinction is easily recognized by both human evaluators and the LLM judge, explaining their high agreement scores.** *Check a more detailed example in post (3).*
>
> See next post.

---

> ### Author Response · Authors · 2025-11-26
> **Response to reviewer brvY - Post(5)**
>
> - **3. Failure of Traditional Metrics**
>   Traditional "content" metrics like CodeBERT yield effectively indistinguishable scores (varying only by ~0.0001) or noisy fluctuations (SBERT). They fail to distinguish between a trivial hardcoded solution and a generalized algorithm, whereas our GEAR-driven approach correlates with genuine human-perceived diversity.
>
> ### **Redefining "Generality" in Abductive Reasoning**
>
> Regarding your concern about the framework's generality due to its reliance on programmable domains, we respectfully suggest that in this nascent field, "generality" should be assessed by the framework's **methodological robustness** rather than its immediate application to every possible modality.
>
> Compared to existing evaluation methods, GEAR introduces three critical properties that establish its generality as a framework:
>
> - **Scalable:** Unlike human-based evaluations which require expensive, non-reproducible manual annotation for every new question, GEAR is label-free and automated.
> - **Reliable:** Unlike textual similarity metrics (e.g., BLEU/ROUGE) which suffer from polysemy, GEAR relies on **functional equivalence**. This aligns with the core philosophy of abduction—that hypotheses are defeasible and should be judged by their predictive power, not just their surface form.
> - **Open-ended:** Traditional benchmarks are static; once an LLM achieves high accuracy, the benchmark "saturates." In contrast, GEAR measures the model's capacity to discover **novel perspectives**. This metric does not saturate; there is always room to generate more diverse, consistent hypotheses.
>
> **A Foundational Step:** While applying GEAR directly to open-domain NLP remains a challenge, this is a limitation of current NLP deduction oracles, not of the GEAR framework itself. GEAR provides the necessary methodology: moving away from static "gold-label matching" in current hypothesis generation research toward dynamic "prediction-pattern analysis." As NLP verification tools mature, the GEAR framework can be naturally extended to those domains.
>
> We believe these additional human and LLM evaluations directly address your core concern: the improvements in our framework are not just syntactic artifacts but represent a genuine, human-verified enhancement in the model's ability to generate diverse, conceptually distinct scientific hypotheses.

---

> ### Author Response · Authors · 2025-11-26
> **Prompt Instruction Used During LLM-Judge and Human Evaluation**
>
> """
> You are a Senior Computer Scientist evaluating the quality and diversity of Python code hypotheses generated for a specific reasoning problem.
>
> Your Goal: Evaluate the set of solutions based on the **User's Problem Description**.
>
> ### Evaluation Criteria (Score 0-10)
>
> 1. Relevance & Creativity (Novelty):
>    - Do the solutions find non-obvious, creative patterns in the data?
>    - Or are they trivial, hard-coded lookups, or simple restatements of the input?
>
> 2. Algorithmic Diversity:
>    - Do the solutions explore *distinctly different* logical paths (e.g., Mathematical vs. String Parsing vs. Object Attributes)?
>    - Or are they minor variations of the same logic?
>
> ### Scoring Guide
>
> - 0-3 (Low):
>   Trivial solutions, hard-coded logic, or all solutions are effectively identical.
>
> - 4-7 (Medium):
>   Valid logic, standard approaches. Some variation if multiple solutions exist.
>
> - 8-10 (High):
>   Highly insightful, novel algorithms that capture the "essence" of the problem. If multiple solutions, they cover the search space broadly.
>
> ### Output Format
>
> Return a JSON object strictly:
> {
>     "reasoning": "Brief analysis of creativity and diversity relative to the problem.",
>     "score": <float 0-10>
> }
>
> ### Problem Description
> {question_text}
>
> ### Generated Solutions
> {code_text}
>
> Evaluate the set of solutions (0-10) based on Algorithmic Diversity and Creativity.
> """

---

> ### Author Response · Authors · 2025-11-28
> **Follow-up on Rebuttal**
>
> Dear Reviewer brvY,
>
> We wanted to send a gentle reminder regarding our latest response. We have provided new human evaluation and LLM-judge experimental results to address your feedback.
>
> With the discussion phase deadline approaching, could you please let us know if these results satisfy your concerns? We are standing by to answer any further questions or additional clarifications if necessary.
>
> Best regards, The Authors

---

### Official Review · Reviewer_MKD8 · 2025-10-30

**Soundness:** 3
**Presentation:** 2
**Contribution:** 1
**Rating:** 2
**Confidence:** 4

**Summary:**

This paper introduces GEAR (General Evaluation for Abductive Reasoning), a new framework for evaluating the abductive reasoning capabilities of Large Language Models (LLMs). The authors argue that existing evaluations, which often rely on a single "gold" hypothesis, are insufficient for abduction. GEAR proposes a label-free, automated evaluation based on three metrics: (1) Consistency (the hypothesis must explain the given observations), (2) Generalizability (the hypothesis should make meaningful predictions on unseen inputs), and (3) Diversity (the model should generate a set of distinct and varied hypotheses). The paper evaluates several LLMs using GEAR on four abduction benchmarks. Additionally, it demonstrates that these metrics can be used as a preference signal to fine-tune LLMs, thereby improving their performance on these same metrics.

**Strengths:**

1. Clarity: The paper is well-written and easy to read.
2. Sensible Criteria: The three proposed criteria for evaluation (Consistency, Generalizability, and Diversity) are sensible, logical, and well-grounded in the principles of scientific reasoning.

**Weaknesses:**

1. Limited Novelty of the Criteria: The primary concern is the limited conceptual novelty of the framework's components. The core metrics proposed are largely a formalization of well-established principles:
     + Consistency (matching observed data) is a fundamental requirement in almost any task.
     + Generalizability (testing on unseen data) is analogous to standard test set evaluation.
     + Diversity (generating distinct solutions) is a known metric in related fields like program synthesis.
The main contribution appears to be the combination of these ideas, rather than a fundamentally new evaluation paradigm.

2. Constrained Domain and Overstated Generality: The framework's key advantages (e.g., being "label-free" and "automated") are heavily dependent on its constrained domain of application. The evaluation is exclusively demonstrated on benchmarks (like MINI-ARC, ACRE) where hypotheses are executable programs. This executable nature is what makes automated checks for Consistency and Generalizability possible; an execution oracle exists. This limits the "general" claim of the framework.
The paper does not solve the core, much harder problem of evaluating open-domain natural language abduction, where ambiguity, logical equivalence, and the lack of an execution oracle are the main challenges.
Furthermore, given the executable setup, using these automated checks as preference signals is a straightforward application of reinforcement learning from feedback. While effective, it feels like an incremental contribution rather than a novel one, especially when the setup itself is so constrained.

**Questions:**

N/A

---

> ### Author Response · Authors · 2025-11-15
> **Response to reviewer MKD8 - Post (1)**
>
> Thank you for your detailed feedback. We agree that our criteria are "sensible and logical," but we respectfully disagree with the assessment of our work's novelty and contribution. We believe this critique stems from a common but incorrect analogy between abductive reasoning and program synthesis (especially under-specified program synthesis problems with NL description or I/O pairs).
>
> ### **On the Novelty of the Criteria (Weakness 1):**
> - **1. Abductive Reasoning is Fundamentally Different from Program Synthesis**
> We respectfully argue that the reviewer's core premise—that our task is analogous to program synthesis and our metrics are "well-established" in that field—is incorrect. The goals of these two tasks are fundamentally different.
>   - **Goal of Program Synthesis:** To **`eliminate ambiguity`**. Given a specification (like I/O pairs), we aim for accuracy and efficiency to solve the problem and correctly implement the user's full intention.
>   - **Goal of Abduction:** To **`embrace ambiguity`**. Given (often sparse) observations, the goal is to generate the broadest possible set of plausible, distinct hypotheses that explain them. ​​We aim for **creativity** and the **broadest possible** hypothesis to enhance future scientific discovery, not simply solve the current 'problem'. The diverse pool of hypotheses is always important in science discovery ([2],[3]) and the difference in their prediction patterns initiated many famous experiments, like Thomas Young's double-slit experiment (testing the 'particle' vs. 'wave' predictions for light) and Eddington's 1919 eclipse observations (testing the quantitatively different predictions of Newtonian gravity vs. Einstein's General Relativity). These experiments not only enhance our understanding of the world, but also boost subsequent new hypothesis propositions.
>
>   - Therefore, metrics for program synthesis are designed to **reward convergence to user's intention**, and abductive reasoning aims for **finding more plausible conceptually alternative explanations**. Our metrics (GEAR) are explicitly designed to reward divergence and explanatory pluralism.
>
>   - **This leads to a final, crucial distinction**: clearer specifications and tighter constraints make program synthesis easier by narrowing the search for the best solution. In contrast, for abductive reasoning, tighter constraints make the task harder, as it demands far greater creativity to find multiple, diverse hypotheses that are consistent with all facts. This confirms they are fundamentally different tasks.(See in fig.3 adding more initial observation (clearer specification) leads to lower score in abduction task)
>
> - **2, The Novelty of GEAR's Metrics**
> Viewed through this lens, our metrics are not *"a formalization of well-established principles"* but rather novel operationalizations for the specific goal of evaluating abductive pluralism.
>   - **Consistency:** We agree this is a fundamental requirement for both tasks.
>   - **Generalizability:** Based on our literature review, 'generalizability' in program synthesis is typically defined as the pass rate on hidden, unseen cases that must match a prediction label. This metric is logical for program synthesis, as its goal is to reward **convergence** to the user's intention. Our Generalizability metric is fundamentally different because our goal is different. We measure **coverage (falsifiability)** over a large, predefined sample space S (with 10,000+ items, versus less than 100 cases in a hidden test case evaluation) and do not have a labeled answer for each test sample point, which is motivated by abduction's goal of discovery. The more such falsifiable cases exist (i.e., have a prediction on a sample point), the more possible experiments we could conduct in the future to verify the hypothesis. The generalizability in abductive reasoning is a concept distinct from the standard generalizability measurement in program synthesis (test accuracy) and one that the field of program synthesis has no motivation to adopt.
>   - **Diversity:** Diversity in program synthesis often refers to syntactic variety (e.g., different ways of achieving the same function to meet the user’s intention). Our $\beta$/$\gamma$ metrics are, to our knowledge, the first to formalize prediction pattern diversity as a direct, objective measure of explanatory and behavioral difference, capturing the creativity prized in abductive reasoning. This is crucial for abduction, where we want conceptually distinct hypotheses. And yet, this very behavior—having various predictions on possible unseen cases—is something traditional program synthesis has no motivation to pursue and **directly contradict to "generalizability" defined in program synthesis tasks** (since it would directly decrease test case accuracy and make the program produce unstable output on unseen cases).
>
>   - Continued in the second post

---

> ### Author Response · Authors · 2025-11-15
> **Response to reviewer MKD8 - Post (2)**
>
> - **Training:** Finally, the reviewer suggests our training is a "straightforward application of RL." We respectfully disagree. The novelty is not in using DPO, but in how we generate the preference signal. We are the first to show that these formal abductive criteria (Consistency, Generalizability, and Diversity) can be used to generate dense, label-free preference pairs, successfully improving abduction without any gold supervision.
>
> The reviewer states these are *"well-established principles,"* but we believe our operationalization of them for label-free abduction is novel. We respectfully ask for any references that formalize Generalizability (as coverage) and Diversity (as prediction pattern dissimilarity) in this way, as our extensive literature review did not find them.
>
>
> ### **On the Constrained Domain and Generality (Weakness 2):**
>
> We agree with the reviewer that our experiments are performed on programmable domains and that open-domain NL abduction is a "core, much harder problem." This was a deliberate methodological choice.
> - As the reviewer notes, NL abduction is challenged by "ambiguity, logical equivalence, and the lack of an execution oracle." We would add that a primary challenge is also objective evaluation. Open-domain NL hypotheses are extremely difficult to evaluate reliably. For example, some benchmarks [1] compare a generated hypothesis to a labeled one using token-level matching. A model might generate, “A carnivorous plant is able to trap insects,” while the label is, “If a plant is carnivorous, then it probably has a trapping structure.” A human sees the semantic similarity, but metrics like ROUGE or BERTScore would fail. Due to this polysemy, it is unreliable to evaluate the true "content" of an NL hypothesis.
>
> - This is precisely why we (and much of the field) use programmable tasks. Code provides a crucial "middle ground": it is far more expressive than simple logic but still permits the objective, deterministic evaluation (via an "execution oracle") that NL currently lacks.
>
> - However, our argument goes a step further: even within programmable tasks, simply evaluating without execution is flawed. Metrics like CodeBERTScore still suffer from polysemy, and structural AST parsing cannot equate a recursive function with an iterative loop that is *functionally identical*.
>
> - Therefore, the current "best practice" and most reliable method is to evaluate hypotheses based on **Functional Equivalence** (i.e., their prediction pattern). This is why programmable tasks are the best choice for this field (we note that ARC is an up-to-date and still challenging benchmark for even today's LLMs), as this approach allows us to robustly capture the core ability of generating novel, plausible hypotheses.
>
> - Our contribution, therefore, is not to have solved the (currently intractable) problem of open-domain NL abduction. Our contribution is to create a general, label-free framework (GEAR) built on this robust foundation of functional equivalence. We are "pointing out a direction" for objective, automated evaluation that does not rely on a single, biased gold label. As we state in our Conclusion (Section 7), this framework can be directly extended to NL once those foundational challenges (like a reliable NL deduction oracle) are met.
>
> - References
>   - [1] Yang, Zonglin, et al. "Language models as inductive reasoners." Proceedings of the 18th Conference of the European Chapter of the Association for Computational Linguistics (Volume 1: Long Papers). 2024.
>   - [2] Thomas C Chamberlin. The method of multiple working hypotheses: With this method the dangers of parental affection for a favorite theory can be circumvented. Science, 148(3671):754–759,1965.
>   - [3]John R Platt. Strong inference: Certain systematic methods of scientific thinking may produce much more rapid progress than others. science, 146(3642):347–353, 1964.

---

> ### Author Response · Authors · 2025-11-19
> **Follow-up on Rebuttal**
>
> Dear Reviewer MKD8,
>
> Hope you're having a productive week.
>
> We're following up on our rebuttal, which we posted four days ago. We hope the clarifications provided successfully addressed your concerns. We would be grateful if you could kindly provide any brief feedback or confirmation regarding whether the main issues are now resolved.
>
> Thanks for your time and guidance on our paper!
>
> Best, The Authors

---

> ### Author Response · Authors · 2025-11-25
> **Follow-up on Rebuttal**
>
> Dear reviewer MKD8,
>
> We hope this message finds you well.
>
> As we approach the end of the discussion period, we wanted to kindly check if you have had a chance to review our detailed responses posted on Nov 15.
>
> We believe our rebuttal directly addresses your primary concerns, particularly regarding the fundamental distinction between Abductive Reasoning and Program Synthesis, and why the metrics of Consistency, Generalizability, and Diversity function differently in the context of scientific discovery versus code generation.
>
> We genuinely value your feedback and would appreciate it if you could reconsider your assessment in light of these clarifications. We remain available to answer any further questions you may have.
>
> Best regards, The Authors

---

> ### Comment · Reviewer_MKD8 · 2025-11-25
> **Response to the Authors**
>
> I thank the authors for their detailed and thoughtful rebuttal. While it clarified the definitions of the metrics and the rationale behind the domain choices, I remain unconvinced regarding the novelty of the proposed framework and the breadth of its contribution.
>
> **1. On Abductive Reasoning and Ambiguity**
> I would like to clarify that I have not conflated Abductive Reasoning with Program Synthesis. I fully appreciate that Abductive Reasoning is inherently a "one-to-many" problem and that embracing ambiguity is central to the task. Utilizing "multiple plausible hypotheses" as a basis for evaluation is a direct application of the consensus definition of abduction, rather than a conceptual breakthrough.
>
> **2. On the Strength of the "Generalizability" Metric**
> I accept your clarification that "Generalizability" is defined here as *coverage* (falsifiability) over a sample space rather than test accuracy. However, I am concerned that this metric is insufficiently discriminative. As shown in your experimental results (e.g., Figure 3(f) and Table 3), the Generalizability scores for most models saturate around 95%–100% for consistent hypotheses. A metric that is so easily saturated offers limited utility for differentiating model capabilities or driving significant improvements.
>
> **3. On the Novelty of Diversity Measures**
> I must respectfully disagree with the assertion that GEAR introduces a novel approach by measuring diversity via "prediction patterns" rather than syntax. Defining diversity based on output disagreement is a cornerstone of **Ensemble Learning**. The proposed metrics map directly onto the taxonomy of diversity measures established in foundational literature, such as Kuncheva & Whitaker (2003) [1], which distinguishes between *pairwise* and *non-pairwise* approaches.
>
> * **Regarding $\gamma$-diversity:** This metric functions as a *non-pairwise* statistic conceptually analogous to the **Entropy measure ($E$)** found in ensemble learning literature. Both are designed to quantify the dispersion or "richness" of the prediction distribution across the hypothesis space. The proposed $\gamma$ can be viewed essentially as a count-based simplification of this classical entropy concept.
>
> * **Regarding $\beta$-diversity:** The formulation of this metric—averaging a dissimilarity score across hypothesis pairs—is structurally equivalent to the **Disagreement measure ($Dis_{av}$)** or the **Q-statistic ($Q_{av}$)** discussed in Kuncheva & Whitaker (2003). The primary distinction lies merely in substituting the binary disagreement count with **Jaccard distance** to accommodate set-based outputs. Consequently, this represents a domain-specific adaptation of the standard *pairwise diversity* framework rather than a novel mathematical definition.
>
> Reference:
> [1] Kuncheva, L. I., & Whitaker, C. J. (2003). Measures of diversity in classifier ensembles and their relationship with the ensemble accuracy. *Machine Learning*, 51, 181-207.
>
> **4. On Domain Constraints**
> I acknowledge the authors' argument that Natural Language (NL) lacks a reliable execution oracle and suffers from ambiguity. However, this ambiguity is precisely the core challenge that a "General" Abductive Reasoning framework needs to address. Bypassing NL in favor of programmable tasks effectively sidesteps the most significant barrier in the field. While the authors state they are "pointing out a direction," I believe a substantial contribution requires tackling these "hard" problems directly, rather than relying on the convenience of an execution oracle which limits the framework's applicability to the very domain where abduction is most critical.
>
> **5. On the Training Strategy**
> Regarding the training contribution, while the specific application to these metrics is new, the methodology itself—generating automated feedback via execution/consistency checks and optimizing via RL—is a direct application of existing paradigms. It represents a logical engineering application of RL from feedback, but does not constitute a methodological innovation in itself.
>
> Given these remaining concerns, I perfer to keep my original score.

---

> ### Author Response · Authors · 2025-11-28
> **Response to Reviewer MKD8 - Post (3)**
>
> Dear Reviewer MKD8,
> Thank you for your prompt reply. We have carefully read your comments and appreciate that you acknowledge the rationale behind our domain choices and the clarification on metric definitions. However, we respectfully wish to address your remaining concerns regarding novelty and contribution, as we believe there are significant distinctions between existing literature and our proposed framework.
>
> - 1. **On Abductive Reasoning and Ambiguity**: We appreciate your confirmation that "Abductive Reasoning is inherently a 'one-to-many' problem" and that "embracing ambiguity is central to the task." **However, we respectfully argue that there is a critical distinction between acknowledging a definition and solving the engineering challenge of evaluating it.**
>   - **The "Operationalization Gap":** While the "one-to-many" nature of abduction is a consensus definition, **no prior work has successfully operationalized this consensus into a scalable and reliable evaluation metric.** As detailed in our literature review, existing benchmarks—even recent ones [1,2,3,4]—still strictly rely on Gold Labels and Gold Answers. They fundamentally treat abduction as a "guess-the-teacher's-password" task, rewarding convergence to a single ground truth rather than rewarding the exploration of the ambiguity you agree is central.
>   - **GEAR’s Unique Contribution:** GEAR is the first framework in the field of Abductive Reasoning to bridge this gap. We provide the first (1) scalable (no human-in-the-loop), (2) reliable (transparent and deterministic), and (3) label-free mechanism to actually measure "multiple plausible hypotheses."
>   - Why This Matters: If we agree that "embracing ambiguity" is the core of the task, then a framework that penalizes deviation from a single gold label (the status quo) is fundamentally flawed. GEAR is the first to fix this alignment issue. **Therefore, if you value the "one-to-many" perspective, we believe it is contradictory to rate this contribution as "poor," as GEAR is the first evaluation metric based on intrinsic hypothesis diversity rather than comparing its similarity to a reference gold answer in the domain of hypothesis generation.**
>
> - 2. **On the Discriminative Power of "Generalizability":** You raised concerns that Generalizability scores saturate (95-100%).
>   - **Reason for High Scores:** This saturation is not a flaw in the metric's logic, but a result of our deliberate choice of Sample Space (S) and the strong instruction-following capabilities of modern LLMs. We chose S to align closely with the original dataset distributions (e.g., keeping entity attributes within the original 0-99 range or specific colors and data type like list and matrix) to ensure fair comparison. Modern models rarely "hard-code" lookup tables when a general logic is apparent, resulting in high coverage.
>   - **Discriminative Potential:** The metric is discriminative if the difficulty is increased. If we expanded S to include out-of-distribution inputs (e.g., negative integers, floats, or unseen colors), the scores would drop. However, even with high scores, this metric serves a crucial purpose: it acts as a necessary gatekeeper. It confirms that a hypothesis is not just a "memorized" look-up table (which would have low coverage) before we credit it for Diversity.
> - 3. **On the Novelty of Diversity Measures (Ensemble Learning vs. GEAR)** We respectfully disagree that our metrics are simply standard Ensemble Learning metrics.
>   - Ecological Origins: Our diversity metrics (β and γ) are grounded in the ecological framework of Whittaker (1960) [5], from which we adapted the nomenclature and mathematical structure.
>   - Distinction from Kuncheva & Whitaker (2003): The work you referenced [6] focuses on binary classifier outputs (correct/incorrect votes) or fixed class labels (0/1).
>     - *Ensemble Learning:* Measures diversity in errors (e.g., "Do classifiers fail on different samples?").
>     - *GEAR:* Measures diversity in semantic behavior (e.g., "Do hypotheses predict different logical outcomes?").
>   - The Contribution: Adapting set-based Jaccard distance to measure the functional equivalence of open-ended hypothesis generation is a novel application. While the mathematical primitives (like Jaccard or Entropy) exist, applying them to prediction patterns of generated programs to quantify explanatory pluralism is new. If we dismiss applied mathematics as lacking novelty, many applied ML contributions would be invalid. We believe bridging ecological/ensemble metrics with abductive hypothesis generation is a valuable methodological contribution.

---

> ### Author Response · Authors · 2025-11-28
> **Response to Reviewer MKD8 - Post (4)**
>
> - 4. **On the "General" Claim and Naming** We understand your concern that "General" might imply solving the AI-complete problem of NL ambiguity.
>   - Clarification: By "General," we meant Benchmark-Agnostic and Scalable (as opposed to human-dependent or dataset-specific metrics), not "Universal for all NL."
>   - Proposed Revision: To avoid misunderstanding and be precise about our scope, we are happy to rename the framework to DEAR: A Diversity-based Evaluation for Abductive Reasoning. We believe this accurately reflects our contribution without overclaiming.
> - 5. **On the Training Strategy** You viewed the training as a logical engineering application of RL.
>   - Novelty of the Reward Signal: Standard RL in reasoning tasks typically rewards convergence to a single gold answer. If we applied standard RL without GEAR, models would collapse to a single mode.
>   - Methodological Innovation: The novelty lies in the objective function, not the optimization algorithm (DPO). We demonstrate that optimizing for Diversity and Consistency (using GEAR's label-free signals) improves the model's fundamental reasoning velocity and transfer accuracy. This is not just "engineering"; it is empirical proof that training for divergence (multiple hypotheses) actually improves accuracy, a non-trivial finding in the context of LLM reasoning.
>
> We have conducted a broad literature review and found no other framework or evaluation metric that:
>   - 1, Evaluates **"multiple plausible hypotheses" without gold labels**.
>   - 2, Is **scalable** and automated (unlike human eval or LLM judge).
>   - 3, Is **open-ended** (improving with discovery rather than saturating at accuracy).
> We believe GEAR (or DEAR) fills a critical gap in the literature. We hope this clarification addresses your concerns regarding the novelty.
>
> References.
> [1] Wang R, Zelikman E, Poesia G, Pu Y, Haber N, Goodman ND. Hypothesis search: Inductive reasoning with language models. arXiv preprint arXiv:2309.05660. 2023 Sep 11.
> [2] Liu E, Neubig G, Andreas J. An incomplete loop: Instruction inference, instruction following, and in-context learning in language models. arXiv preprint arXiv:2404.03028. 2024 Apr 3.
> [3] Bowen C, Sætre R, Miyao Y. A comprehensive evaluation of inductive reasoning capabilities and problem solving in large language models. InFindings of the Association for Computational Linguistics: EACL 2024 2024 Mar (pp. 323-339).
> [4] He K, Chen Z. From reasoning to learning: A survey on hypothesis discovery and rule learning with large language models. Transactions on Machine Learning Research. 2025.
> [5] Whittaker, R. H. "Vegetation of the Siskiyou mountains, Oregon and California." Ecological Monographs 30.3 (1960): 279-338.
> [6] Kuncheva, L. I., & Whitaker, C. J. "Measures of diversity in classifier ensembles." Machine Learning 51 (2003): 181-207.

---

### Official Review · Reviewer_9oVs · 2025-10-31

**Soundness:** 3
**Presentation:** 2
**Contribution:** 3
**Rating:** 4
**Confidence:** 2

**Summary:**

This paper highlights that many existing benchmarks for abductive reasoning rely on a single annotated “gold” hypothesis, which is limiting because multiple explanations may be equally valid. Rather than introducing new datasets, the framework described in this paper (GEAR) restructures several existing benchmarks — MINI-ARC, ARC-2025, LIST FUNCTIONS, and ACRE to evaluate the consistency, generalizability, and diversity of candidate hypotheses. While the framework is conceptually clear, it is difficult to follow precisely how the the restructuring is implemented. In particular, it appears that instances (or existing groups of tasks?) are grouped to test individual hypotheses across multiple observations, but the specifics are not fully described. Moreover, it is unclear how textual benchmarks such as ACRE are evaluated for alternative plausible hypotheses, leaving some ambiguity about the method’s applicability to natural-language domains.

**Strengths:**

- Thoughtful reformulation of existing tasks to allow evaluation of multiple plausible hypotheses.
- Evaluation across multiple large language models

**Weaknesses:**

- Several parts of the paper are unclear, particularly the construction of the evaluation datasets and the specifics of model evaluation.
- The computation of metrics such as Instruction-Following Rate is not clearly defined.

**Questions:**

1. Could you provide more details or examples of how the evaluation datasets were constructed for each benchmark, including how examples were grouped and how synthetic or additional test cases were generated? How does one ensure that the grouping is always feasible and sound?

2. Could you clarify the training procedure — specifically, how the GEAR scores were used during training and how evaluation on unseen portions of the original tasks was performed?

3. How exactly was the Instruction-Following Rate in Table 2 computed, and how does it interact with other metrics?

4. Were metrics from the original benchmarks used in any way, or is evaluation entirely based on GEAR’s framework?

Overall, I find the conceptual contribution promising, but the lack of clarity in critical implementation details makes it difficult to fully assess the experimental results. I look forward to clarifications during the author discussion period and will revise my review accordingly.

---

> ### Author Response · Authors · 2025-11-15
> **Response to reviewer 9oVs - Post (1)**
>
> Thank you for your thoughtful comments and suggestions. We would like to address your concerns as follows:
> ### **Response to Weakness 1:**
>
> - We apologize for the confusion; this section was compressed due to page limits. As stated in Section 4.1, we use publicly available abductive reasoning datasets: ACRE, LIST FUNCTIONS, ARC, and MINI-ARC. Originally, these datasets are split into  $\mathbb{O}\_{train}$ and $\mathbb{O}\_{test}$, where a model is evaluated on its ability to form a hypothesis from $\mathbb{O}
> _{train}$ that correctly predicts $\mathbb{O}\_{test}$.
> - However, as we argue in Sections 1 and 2, these splits are often derived from a single "gold-labeled" hypothesis, which introduces bias. Given the defeasible nature of abductive reasoning, we argue that any plausible hypothesis that correctly explains the given observations should be considered sound. As we illustrated in Figure 1, a fixed set of initial observation can often be explained by far more hypotheses than just the single ‘gold’ one used to create the original test split. Therefore, we did not use the original $\mathbb{O}\_{train}$ and $\mathbb{O}\_{test}$ split.
> - For example, in the original LIST FUNCTIONS benchmark, a problem might be:
>   - $\mathbb{O}\_{train}$ = `[[1,2,3,4] -> [1,3], [3,4,5,6]->[3,5], [4,5,6,7]->[5,7]]` (3 observations)
>   - $\mathbb{O}\_{test}$ = `[[5,6,7,8]-> [5,7]]` (1 observation)
> - In our evaluation, these are combined into $\mathbb{O}\_{all}$, which includes 4 observations in total. We then test the model's ability given different numbers of initial observations by sampling combinations from this set. This $\mathbb{O}\_{all}$ (with 4 observations) can be used to construct 15 distinct problems with $n \in \{1, 2, 3, 4\}$ initial observations ($C^4\_1 + C^4\_2 + C^4\_3 + C^4\_4 = 15$). During our experiments, we use a stratified sampling method to ensure the number of evaluation problems is balanced across these different counts of initial observations.
> - Since our setting is the combination of train and original test, then at least it has one solution (the original gold hypothesis). Then we sample the subset which makes it looser, so it **must be feasible**. When more initial observations are provided, the constraints become tighter, making it more challenging for the LLM to generate new novel perspectives but still there will be at least a feasible solution .
>
> ### **Response to Weakness 2:**
> - Since instruction following is not the main focus of our work on abductive reasoning, we only discussed it briefly. When we get a generated string from an LLM, we must parse it into an executable function to run our evaluation. However, some lower-performing models generated responses can not be parsed by python interpreter, wrong in syntax and can not be executed. We refer to this as a failure to follow instructions, or "non-parsable."
>
> ### **Response to Question 1:**
> - Please see our response to Weakness 1.
>
> ### **Response to Question 2:**
>
> - We use DPO (an offline RL training method), so our GEAR metrics are used to construct preference pairs before training begins. Given a set of initial observations $\mathbb{O}\_{init}$​, a context of existing hypotheses $H$, and two new candidate hypotheses $h\_1$​ and $h\_2$​, the preference construction follows a 3-stage hierarchy. First, we test for **Instruction Following**: we always prefer a hypothesis that is instruction-following (i.e., executable and parsable code) over a non-instruction-following one. Second, if both hypotheses are instruction-following, we test for **Consistency**: we prefer the hypothesis that is consistent with $\mathbb{O}\_{init}$​ over an inconsistent one. Third, if both hypotheses are consistent, we evaluate them with **GEAR**: as detailed in Section 5.3, the final **GEAR score** is the average of three dimensions: (1) Generalizability, (2) $\gamma$ diversity, and (3) $\beta$ diversity. The candidate hypothesis with the higher GEAR score is labeled as 'preferred,' and the other is labeled as 'rejected."
>
> - We chose DPO because it is efficient and we had already generated a large number of candidate hypotheses. GEAR can also be used in online RL training (like GRPO); simply replacing the reward signal with the GEAR score would work.
>
> - The evaluation on unseen problems follows the same procedure from Section 5 (the problems are held-out from training). Using the example from Weakness 1, if evaluating with 1 initial observation, the LLM sees $o$ = [1,2,3,4] -> [1,3] and is asked to generate 3 different hypotheses iteratively: first $h\_1$, then $h\_2$ (given $o$ and $h\_1$), and finally $h\_3$ (given $o$, $h\_1$, and $h\_2$).

---

> ### Author Response · Authors · 2025-11-15
> **Response to reviewer 9oVs - Post (2)**
>
> ### **Response to Qeustion 3:**
> - It is simply the rate of generated strings that can be successfully parsed into a runnable Python function. If the code has a syntax error, or the string does not follow the required output format, it is counted as not following instructions. In our prompt (Appendix G), we require the LLM to generate a tuple: ("Natural language explanation", "`def f(x): ...`"). If an LLM fails to generate a parsable function, its diversity scores will be low. For example, if only 1 out of 3 generated hypotheses can be parsed, the $\gamma$ score for that problem will be 1 (as there is only one valid hypothesis contributing to the prediction space).
>
> ### **Response to Question 4:**
> - The metric from the original benchmarks is typically a pass/fail accuracy based on the single "gold" hypothesis's predictions on unseen test cases. As we argue, this is a biased evaluation, since a logically sound abductive hypothesis may still differ from the gold answer on unseen data. Therefore, we do not use this metric directly.
> - However, we show in Table 2 that how original evaluation metric is related to ours. When evaluating a set of hypotheses, an LLM that generates a more diverse set (higher GEAR score) also achieves a higher Top-N accuracy (i.e., the probability that at least one hypothesis in the set of $n$ generated hypotheses matches the gold answer's predictions). The intuition is straightforward: by encouraging the model to generate hypotheses based on different mechanisms, hypothesis set with more mechanism increases the chance that one of them will "hit" the gold-labeled one.
>
> ### **We would also like to share a qualitative example showing the effectiveness of GEAR-derived DPO training**
>
> - **Problem:** (Given 6 input/output pairs, Pair0: (['blue rubber cylinder', 'green rubber cube'], on), Pair1: (['blue rubber cylinder'], on), Pair2: (['green rubber cube'], off), Pair3: (['gray rubber cylinder'], off), Pair4: (['cyan rubber cube'], on), Pair5: (['brown metal sphere', 'red rubber sphere'], off))
> - Generated Hypotheses (**BEFORE** Training):
>   - **H1:** ("Turn on if there is a blue rubber cylinder, otherwise turn off", `def f(x):\n if 'blue rubber cylinder' in x:\n return 'on'\n return 'off'`)
>   - **H2:** ("Turn on if the count of 'blue rubber cylinder' is odd, otherwise turn off", `def f(x):\n count = x.count('blue rubber cylinder')\n return 'on' if count % 2 == 1 else 'off'`)
>   - **H3:** ("Turn on if any object is 'blue rubber cylinder', otherwise turn off", `def f(x):\n return 'on' if 'blue rubber cylinder' in x else 'off'`)
> - Generated Hypotheses (**AFTER** Training):
>   - **G1:** ("Return 'on' if the list contains 'blue rubber cylinder' or 'cyan rubber cube', else 'off'", `def f(x):\n if 'blue rubber cylinder' in x or 'cyan rubber cube' in x:\n return 'on'\n return 'off'`)
>   - **G2:** ("Return 'on' if the number of items in the list is odd, else 'off'", `def f(x):\n if len(x) % 2 == 1:\n return 'on'\n return 'off'`)
>   - **G3:** ("Return 'on' if any color starts with 'b', 'c', or 'g' (irrespective of material), else 'off'", `def f(x):\n colors = [item.split()[0] for item in x]\n if any(color in ['blue', 'brown', 'cyan', 'green'] for color in colors):\n return 'on'\n return 'off'`)
> - As you can see, the Before Training set (H1-H3) is not very diverse; all three hypotheses are minor variations focused only on the single 'blue rubber cylinder' feature. H1, H2, and H3 are largely functionally identical. The After Training set (G1-G3) is far more conceptually diverse. G2 explores a meta-feature (the list's length, `len(x)`), and G3 explores a completely novel explanatory angle (abstract color properties, `item.split()[0]`) that the base model did not find.
> - **Most importantly, this increase in novelty was accompanied by a dramatic improvement in consistency.** The Before Training hypotheses (H1-H3), being largely functionally identical, are all inconsistent with the full set of observations. In contrast, the After Training set, by exploring new perspectives, successfully finds a fully consistent hypothesis (G1) that correctly explains all 6 cases. This shows the LLM learned to find a correct rule, not just a different one. This dual improvement is reflected in the scores for this single problem:
> - **Consistency (Avg. Cases Matched):**
>   - Before: 0 / 3 hypotheses are consistent.
>   - After: 1 / 3 hypotheses is fully consistent (G1).
> - **Diversity (Total Unique Mappings):**
>   - Before: 7,049 ($H2$ and $H3$ are identical so every hypothesis have identical prediction on each sample point.)
>   - After: 13,206

---

> ### Author Response · Authors · 2025-11-19
> **Follow-up on Rebuttal**
>
> Dear Reviewer 9oVs,
>
> Hope you're having a productive week.
>
> We're following up on our rebuttal, which we posted four days ago. We hope the clarifications provided successfully addressed your concerns. We would be grateful if you could kindly provide any brief feedback or confirmation regarding whether the main issues are now resolved.
>
> Thanks for your time and guidance on our paper!
>
> Best,
> The Authors

---

> > ### Comment · Reviewer_9oVs · 2025-11-20
> > **Acknowledgment Of Rebuttal**
> >
> > Hi, Thank you for you response - I wanted to let you know that I have noted the comments. I'll need time to take a detailed look and I plan to do that on Friday. Will update the review/follow up accordingly.

---

> > > ### Comment · Reviewer_9oVs · 2025-11-22
> > > **Update after rebuttal**
> > >
> > > Thanks - I think these are very crucial details and should be included in the paper. I am updating my score for the paper.

---

### Official Review · Reviewer_5GLN · 2025-10-31

**Soundness:** 4
**Presentation:** 4
**Contribution:** 3
**Rating:** 8
**Confidence:** 4

**Summary:**

The paper presents GEAR, a label-free framework for evaluating abductive reasoning in LLMs. It assesses sets of hypotheses using three interpretable criteria: (1) Consistency with observations, (2) Generalizability across a defined input space, and (3) Diversity among alternative explanations. The authors applied GEAR to four code-based abductive benchmarks and nine LLMs. It reveals that even large models struggle with consistency and produce limited diversity. The authors further use GEAR as a training signal through a momentum-based curriculum DPO approach, improving both diversity and downstream abductive accuracy.

**Strengths:**

Originality: The authors introduced a principled, label-free evaluation of abductive reasoning grounded in philosophy of science.

Significance: The paper addresses a fundamental gap in assessing LLMs’ capacity to generate plausible, falsifiable hypotheses.

Quality: Experiments are extensive (9 LLMs, 50K hypotheses) and include insightful analyses, e.g., defeasibility and the correlation between GEAR scores and generalization.

Clarity: Well-written with precise definitions and clear visualisations. The momentum-based curriculum is simple yet effective.

**Weaknesses:**

1. The framework depends on a predefined sample space and deterministic execution, limiting use beyond structured domains.

2. Training evaluation omits comparisons to other prompting baselines.

3. Heuristic design choices (e.g., 3-error stopping rule, sample-space construction) lack sensitivity analysis.

4. The interpretability of diversity metrics (γ, β) could be improved with qualitative examples.

**Questions:**

1. Why does model size not correlate with abductive diversity—does pretraining or alignment explain this?

2. Could you elaborate on Line 415 (“some benefit from earlier emphasis on format/consistency, others from earlier diversity”)? And can you provide qualitative examples illustrating what “high-diversity” vs. “low-diversity” hypotheses look like?

3. How could GEAR extend to natural-language abduction tasks beyond formal domains?

---

> ### Author Response · Authors · 2025-11-15
> **Response to reviewer 5GLN - Post (1)**
>
> Thank you for your very thoughtful review and for your strong support of our paper. We are grateful for your positive feedback on the originality, significance, and quality of our work, and we appreciate that you found the paper well-written and the definitions precise.
>
> We would like to address your feedback and questions below.
>
> ### **Response to Weakness 1 & Question 3 (Domain Limitation & NL Extension):**
> Thank you for this point. We agree that the current framework is demonstrated on executable domains. We would like to clarify two key points:
> - **Comparable Evaluation Effort:** As you noted, our framework requires a predefined sample space S. We argue that this effort is comparable to, not greater than, the effort required for constructing traditional "gold-label" benchmarks. Those benchmarks require expensive, manual annotation of a single "gold" hypothesis and a set of curated test cases for every single problem. In contrast, S can often be defined once for an entire domain (e.g., all ACRE problems) and be reused, making it a different, and often more scalable, type of effort.
> - **Path to Natural Language:** As we discuss in our Conclusion (Section 7), the principles of GEAR (Consistency, Generalizability, Diversity) are fully general. The primary bottleneck for extending GEAR to open-domain NL is the current lack of two foundational tools: (1) a reliable "NL deduction oracle" to test a hypothesis, and (2) a robust "semantic equivalence predicate" to handle polysemy (i.e., knowing that "A carnivorous plant traps bugs" and "Plants that eat insects have traps" are functionally similar). Our framework 'points the way' for evaluation, and as these foundational NL tools become more reliable, GEAR's principles can be directly adopted.
>
> ### **Response to Weakness 2 (Prompting Baselines):**
> - We intentionally used a minimal, dataset-agnostic prompt to create a fair and controlled comparison. Our goal was to purely evaluate the core hypothesis generation (abductive) step for various LLMs.
> - Many advanced prompting methods (e.g., self-reflection, ReAct, or iterative refinement) are powerful but blend abduction with deductive verification and inductive refinement. Using such complex pipelines would make it difficult to isolate and evaluate the core abductive-reasoning ability, which was our primary objective. We believe our simple prompt serves as the most effective and "pure" baseline for this specific task.
>
> ### **Response to Weakness 3 (Heuristics / Sensitivity Analysis):**
> - You are correct that choices like the "3-error stopping rule" are heuristics. A full sensitivity analysis (e.g., testing stopping rules of 1, 3, 5, 10 errors, or different methods for constructing S) would be computationally very expensive, given the 9 LLMs and 1,500 problems. However, we believe our chosen settings are effective for the paper's goal: they are generous enough to allow stronger models to demonstrate their full capabilities (as seen with GPT-o1) while still being strict enough to differentiate them from weaker models. The clear and consistent performance gaps in Figure 3 suggest these heuristics are successful at revealing meaningful differences.

---

> ### Author Response · Authors · 2025-11-15
> **Response to reviewer 5GLN - Post (2)**
>
> ### **Response to Question 1 (Model Size vs. Diversity):**
> While we can only hypothesize, we believe this may be due to two primary factors
> - **Pretraining Objective:** Standard pre-training (next-token prediction) excels at learning common patterns, facts, and "mean" human responses. It does not explicitly train for the creative, counter-factual, "what-if" reasoning required to generate multiple, novel, and distinct hypotheses.
> - **Standard Alignment (RLHF):** Many alignment procedures (like RLHF) optimize for a single, safe, helpful, and widely-accepted response. This process could inadvertently penalize "unusual," "creative," answers, effectively suppressing the model's ability to engage in the pluralistic reasoning that abduction requires.
> We believe this finding is significant because it suggests abductive reasoning is a distinct capability that cannot be taken for granted, even in large models. It must be explicitly evaluated and, as our training experiments show, explicitly trained for. This is the core motivation for GEAR.
>
> ### **Response to Weakness 4 & Question 2 (Qualitative Examples & Line 415):**
> - **On Line 415:** This refers to our observation during initial experiments. We found that different base models had different paths to improvement. For example:
>   - **Llama-3.1-8B** struggled heavily with basic instruction-following (see its very low 0.149 rate in Table 2 / Fig 3b). For it, the 'easiest' and fastest-improving signal was "Parsing."
>   - **Qwen-2.5-7B**, in contrast, was already highly compliant (0.988 rate). For it, the "Parsing" signal was already 'solved,' and it could immediately start learning from the "Consistency" signal.
>   - Our momentum curriculum automatically detects this "learning velocity." It adaptively feeds Llama the parsing-preference pairs it desperately needs, while allowing Qwen to "move on" to the harder consistency and diversity signals. This avoids an inefficient, one-size-fits-all curriculum.
>
> - **On Qualitative Examples:**
> You are right, qualitative examples are key to understanding β/γ diversity. Here is an excellent example from the ACRE dataset (ID 66) for the Nextcoder-7B model, showing the training effectiveness.
> - **Problem:** (Given 6 input/output pairs, Pair0: (['blue rubber cylinder', 'green rubber cube'], on), Pair1: (['blue rubber cylinder'], on), Pair2: (['green rubber cube'], off), Pair3: (['gray rubber cylinder'], off), Pair4: (['cyan rubber cube'], on), Pair5: (['brown metal sphere', 'red rubber sphere'], off))
> - Generated Hypotheses (**BEFORE** Training):
>   - **H1:** ("Turn on if there is a blue rubber cylinder, otherwise turn off", `def f(x):\n if 'blue rubber cylinder' in x:\n return 'on'\n return 'off'`)
>   - **H2:** ("Turn on if the count of 'blue rubber cylinder' is odd, otherwise turn off", `def f(x):\n count = x.count('blue rubber cylinder')\n return 'on' if count % 2 == 1 else 'off'`)
>   - **H3:** ("Turn on if any object is 'blue rubber cylinder', otherwise turn off", `def f(x):\n return 'on' if 'blue rubber cylinder' in x else 'off'`)
> - Generated Hypotheses (**AFTER** Training):
>   - **G1:** ("Return 'on' if the list contains 'blue rubber cylinder' or 'cyan rubber cube', else 'off'", `def f(x):\n if 'blue rubber cylinder' in x or 'cyan rubber cube' in x:\n return 'on'\n return 'off'`)
>   - **G2:** ("Return 'on' if the number of items in the list is odd, else 'off'", `def f(x):\n if len(x) % 2 == 1:\n return 'on'\n return 'off'`)
>   - **G3:** ("Return 'on' if any color starts with 'b', 'c', or 'g' (irrespective of material), else 'off'", `def f(x):\n colors = [item.split()[0] for item in x]\n if any(color in ['blue', 'brown', 'cyan', 'green'] for color in colors):\n return 'on'\n return 'off'`)
> - As you can see, the Before Training set (H1-H3) is not very diverse; all three hypotheses are minor variations focused only on the single 'blue rubber cylinder' feature. H1, H2, and H3 are largely functionally identical. The After Training set (G1-G3) is far more conceptually diverse. G2 explores a meta-feature (the list's length, `len(x)`), and G3 explores a completely novel explanatory angle (abstract color properties, `item.split()[0]`) that the base model did not find.
> - **Most importantly, this increase in novelty was accompanied by a dramatic improvement in consistency.** The Before Training hypotheses (H1-H3), being largely functionally identical, are all inconsistent with the full set of observations. In contrast, the After Training set, by exploring new perspectives, successfully finds a fully consistent hypothesis (G1) that correctly explains all 6 cases. This shows the LLM learned to find a correct rule, not just a different one. This dual improvement is reflected in the scores for this single problem:
> - **Consistency (Avg. Cases Matched):**
>   - Before: 0 / 3 hypotheses are consistent.
>   - After: 1 / 3 hypotheses is fully consistent (G1).
> - **Diversity (Total Unique Mappings):**
>   - Before: 7,049
>   - After: 13,206

---

> > ### Comment · Reviewer_5GLN · 2025-11-25
> >
> > Thanks for your response and clarifications to my concerns and questions. My points regarding prompting baselines and qualitative examples have been addressed, though it remains a question how GEAR can be extended to natural-language abduction tasks. Despite this limitation, I am still positive about the contribution of the paper and hence keep my score.

---

> ### Author Response · Authors · 2025-11-19
> **Follow-up on Rebuttal**
>
> Dear Reviewer 5GLN,
>
> Hope you're having a productive week.
>
> We're following up on our rebuttal, which we posted four days ago. We hope the clarifications provided successfully addressed your concerns. We would be grateful if you could kindly provide any brief feedback or confirmation regarding whether the main issues are now resolved.
>
> Thanks for your time and guidance on our paper!
>
> Best,
> The Authors

---

### Author Response · Authors · 2025-11-15
**To all Reviewers**

Dear Reviewers,

- We sincerely thank all of you for your time and for providing such thorough, high-quality reviews of our paper. Your thoughtful comments, challenging questions, and constructive criticisms have been invaluable to us. We appreciate the deep engagement with our work.

- We have posted detailed, individual responses to each review (5GLN, 9oVs, MKD8, and brvY) to address every point raised, from clarifying our experimental setup and metrics to discussing the core conceptual framing of our contribution.

- We hope these responses successfully address your concerns and clarify the novelty and significance of GEAR.  If any of our answers are unclear, or if new questions arise, we would be more than happy to dive into a more in-depth discussion.

Thank you once again for your feedback!

---

### Meta-Review · Area_Chair_sPqo · 2026-01-07

**Summary:**

The paper proposes GEAR, a label-free framework to evaluate abductive reasoning in LLMs by scoring sets of hypotheses along three axes: consistency with observations, generalizability over a predefined sample space, and diversity among hypotheses. It applies GEAR to several executable, code-based abduction benchmarks across multiple LLMs, finding that even large models often lack consistency and produce low diversity. The authors also use GEAR-derived preferences in a curriculum-style DPO fine-tuning setup and report improvements in diversity and some downstream abductive accuracy.

Pros
1. Clear motivation: single “gold” hypotheses are a poor fit for underdetermined abduction; evaluating a set of hypotheses is sensible.
2. Well-structured framework with interpretable dimensions (consistency/generalizability/diversity) and large empirical sweep across models/datasets.
3. The training study is pragmatic and shows GEAR can be turned into an automated feedback signal, with qualitative examples suggesting the curriculum can reduce mode collapse and broaden hypothesis generation.

Cons
1. Generality is overstated: the framework relies heavily on an executable oracle and a hand-defined sample space; extension to open-domain natural language abduction remains speculative and unresolved.
2. Key components feel incremental: the three criteria align closely with standard desiderata, and the diversity measures largely operationalize output-disagreement/coverage ideas that are well-studied in adjacent areas; novelty is mainly in packaging and applying them to these benchmarks.
3. Sensitivity/robustness is not fully established: multiple heuristics (sample-space construction, stopping rules, evaluation setup) could materially affect outcomes, but the analysis is limited.
4. Impact of training is mixed: gains on some metrics appear modest or hard to interpret without deeper ablations on why the curriculum helps and when it transfers beyond the GEAR-defined setting.
5. Some clarity issues (raised by reviewers) suggest important implementation details were initially under-specified, which makes it harder to fully trust or reproduce the evaluation pipeline.

While the work is careful and the problem framing is valuable, the submission’s central claims (a “general” label-free evaluation for abduction) are not convincingly supported beyond executable, synthetic/programmatic domains where an oracle exists. The rebuttal improves clarity and adds qualitative/human/LLM-judge evidence that the diversity signal correlates with perceived conceptual variation, but it does not fully address the core concern that the framework’s applicability and novelty are limited by dependence on curated sample spaces and functional execution. Overall, I see this as a solid empirical/system-building contribution with promising direction, but not yet strong enough in generality, methodological novelty, or demonstrated transfer to warrant acceptance at this venue.

**Reviewer Concerns:**

1. Prompting baselines missing: Authors justify using a minimal, dataset-agnostic prompt to isolate “pure” hypothesis generation, and argue more complex prompting would confound abduction with verification/refinement. Reviewer 5GLN explicitly agrees this was addressed.
2. Need qualitative examples for diversity metrics γ/β: Authors add concrete before/after examples showing broader hypothesis families post-training; brvY also gets an added human + LLM-judge study claiming correlation between prediction-pattern diversity and “conceptual” diversity.
3. Clarity of dataset restructuring / how evaluation instances are formed (9oVs): Rebuttal explains how they merge original train/test observations into a set and sample subsets by number of initial observations; 9oVs indicates these details were crucial and updated their score upward.
4. How training uses GEAR: Authors clarify preference construction (parsing > consistency > GEAR) and the offline DPO setup; this directly answers the “how is it used” question.
5. Instruction-following rate definition: Authors define it as “parsable/executable code rate,” resolving the metric definition confusion.

**Reviewer Scores:**

N/A

---

### Decision · Program_Chairs · 2026-01-26

Reject